# The twist-and-squeeze activation of CARF-fused adenosine deaminase by cyclic oligoadenylates

Charlisa Whyms[1,6,8], Yu Zhao [ID][2,8], Doreen Addo-Yobo [ID][1,8], Huan He[3], Arthur Carl Whittington[4], Despoina Trasanidou[5,7], Carl Raymund P Salazar [ID][5], Raymond H J Staals[5] & Hong Li [ID][2✉]

## Abstract

**The recently identified CARF (CRISPR-associated Rossman-fold) family of proteins play a critical role in prokaryotic defense, mediating cOA (cyclic oligoadenylate)-stimulated ancillary immune responses in the type III CRISPR-Cas systems. Whereas most previously characterized CARF proteins contain nucleic acids or protein degradation effectors, a subset of the family, including the CARF-fused adenosine deaminase (ADA) (Cad1), has recently been shown to convert ATP to ITP. The enzymatic mechanism and the activation process of Cad1, however, remain incompletely understood. Here we present biochemical and structural analyses of a ring nuclease Cad1, revealing its substrate binding specificity and a sequential activation process by cOAs. Despite an overall structural similarity to canonical ADA enzymes, the ADA domain of Cad1 possesses unique structural features that confer a specificity for ATP. Supported by mutational analysis, our structural work demonstrates an allosteric link between the cOA-binding CARF and the ADA domain through a protein network within the hexameric enzyme assembly. Binding of a cA4 molecule to paired CARF domains induces a twisting of the linked ADA domains around one another, which remodels their active sites and alters interactions with neighboring ADA domains, thereby driving a sequential conformational activation mechanism.**

**Keywords** Adenosine Deaminase; CRISPR-Associated Rossmann Fold; Cyclic Oligoadenylate; Sequential Model of Allostery
**Subject Categories** Microbiology, Virology & Host Pathogen Interaction; Structural Biology

## Introduction

Virtually all domains of life utilize linear or cyclic nucleotide-based second messengers to regulate diverse cellular processes in response to external or internal stimuli (Pesavento and Hengge, 2009).

Recently, the cyclic oligonucleotide (cO)-based signaling strategy has been expanding to prokaryotes in the CRISPR (clustered regularly interspaced short palindromic repeats)-Cas (CRISPR-associated) systems. CRISPR-Cas confers adaptive immunity to its hosts against the viruses and other invasive mobile genetic elements (Barrangou et al, 2007; Brouns et al, 2008; Marraffini and Sontheimer, 2010). The Type III CRISPR-Cas systems exemplify this functionality by eliciting multi-pronged and multi-leveled antiviral responses. The Type III CRISPR-Cas effector complex senses, via a protein-associated CRISPR RNA (guide RNA), and degrades the viral transcripts (target RNA). Binding of the target RNA triggers two enzymatic activities within its Cas10 subunit: the nonspecific shredding of single-stranded DNA and the synthesis of cOA molecules from adenosine triphosphate (ATP) (Kazlauskiene et al, 2017; Niewoehner et al, 2017). The cOA molecules subsequently stimulate the unique CARF (CRISPR-associated Rossman-fold) domain-containing family of proteins that further assist with clearing the infection. An analogous process occurs in the CBASS (cyclic oligonucleotide-based-anti-phage-signaling-system) systems in which phage infection also stimulates the production of diverse cyclic oligonucleotides that then activate downstream proteins, ultimately halting the propagation of the phages (Athukoralage and White, 2022; Millman et al, 2020; Patel et al, 2022). The cOA-mediated signaling processes provide a wealth of insights into enzyme regulation principles that benefit both basic research and biotechnology development.

Bioinformatic analysis has unveiled an impressive collection of diverse CARF proteins that are made of the CARF domain alone or a fused effector domain of various enzymatic functions (Makarova et al, 2014). The CARF family of proteins have been classified into 9 major and 13 minor clades (Makarova et al, 2020). The most common CARF proteins associated with the Type III CRISPR-Cas systems are the CARF-fused HEPN (higher eukaryotes and prokaryotes nucleotide-binding) or PD/D/ExK nucleases such as Csm6/Csx1 and the CARD1-like proteins (Du et al, 2024; Kazlauskiene et al, 2017; Niewoehner et al, 2017; Niewoehner and Jinek, 2016; Rostol et al, 2021). Structural and biochemical analysis reveal that these CARF-nucleases bind specific cOA molecules that allosterically enhance the catalytic activity of their fused effector domains (Athukoralage and White, 2022) by diverse mechanisms

[1]Department of Chemistry and Biochemistry, Florida State University, Tallahassee, FL 32306, USA. [2]Department of Structural Biology, Van Andel Institute, Grand Rapids, MI 49503, USA. [3]Institute of Molecular Biophysics, Florida State University, Tallahassee, FL 32306, USA. [4]Department of Biological Sciences, Florida State University, Tallahassee, FL 32306, USA. [5]Laboratory of Microbiology, Wageningen University and Research, Stippeneng 4, 6708WE Wageningen, The Netherlands. [6]Present address: Chemistry Department, United States Naval Academy, Annapolis, MD 21402, USA. [7]Present address: Sanquin, Postbus 9892, 1006 AN Amsterdam, The Netherlands. [8]These authors contributed equally: Charlisa Whyms, Yu Zhao, Doreen Addo-Yobo. ✉E-mail: hong.li@vai.org

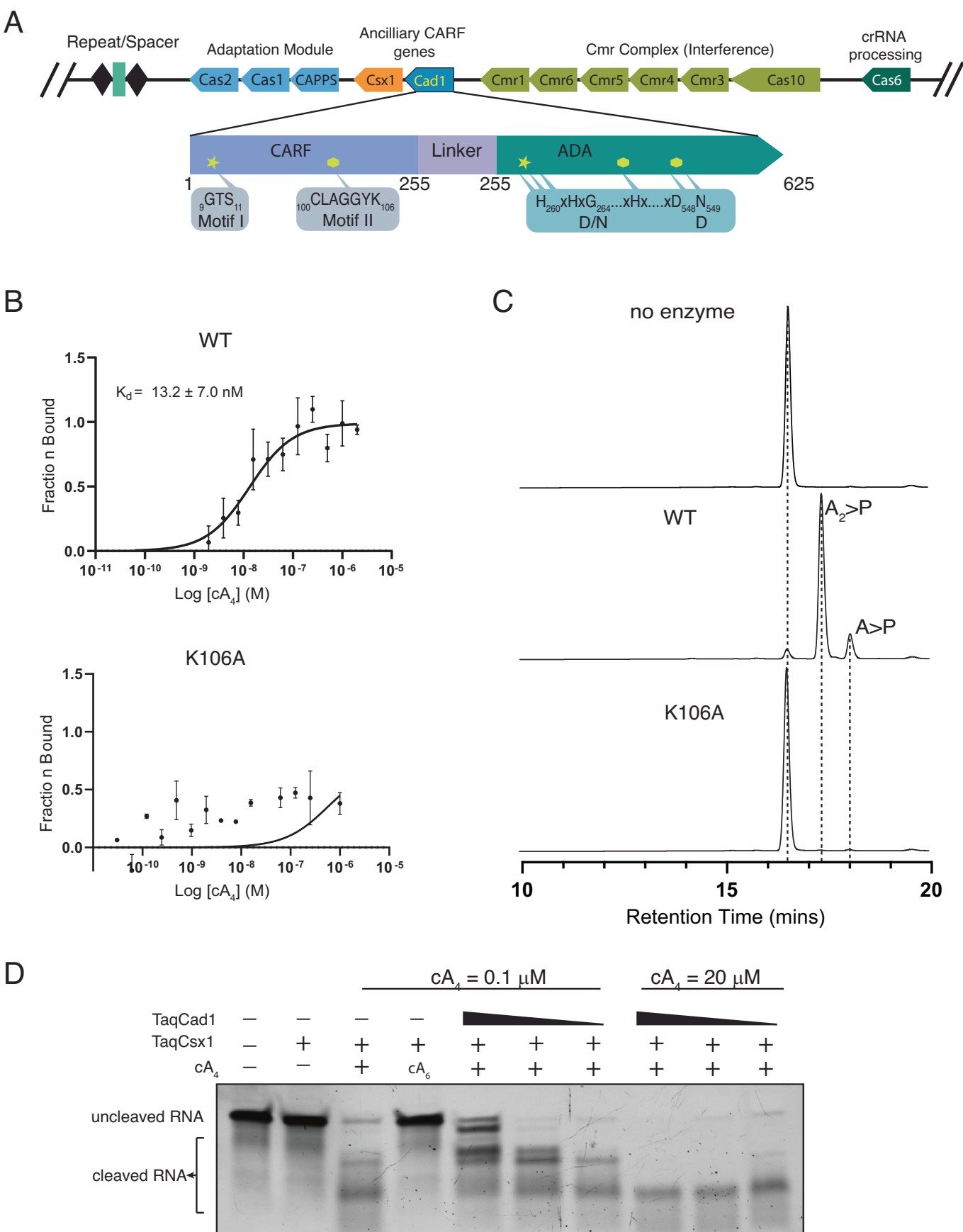

**Figure 1.   *Taq*Cad1 features and biochemical activities.**

(A) *Thermoanaerobaculum aquaticum* CRISPR locus and TaqCad1 sequence features. Key amino acids along with their residue numbers for the two enzymatic domains are shown in the insets, where "D/N" and "D" describe the residues conserved in canonical adenosine deaminase active sites. (B) Microscale thermophoresis (MST) titration curves of cyclic tetra-adenosine (cA$_4$) to *Taq*Cad1 or its mutant K106A. The center of each data point represents the mean, and the error bar represents the standard deviation of the mean from $n = 3$ biological replicates. The apparent $K_d$ is estimated by fitting the binding isotherm to a one-specific binding model. No data fitting was attempted for the K106A titration. (C) Elution profiles of cA$_4$ and its hydrolyzed products on HPLC by *Taq*Cad1 or K106A. (D) Activation of *Taq*Csx1 by cA$_4$ and the impact of *Taq*Cad1 ring nuclease activity on *Taq*Csx1 RNA cleavage activity at two different cA$_4$ concentrations. Source data are available online for this figure.

(Athukoralage and White, 2021; Das et al, 2022), ranging from controlling the catalytic metal ions (Rostol et al, 2021; Zhu et al, 2021) to oligomerization (Molina et al, 2019) of the effector domains. While the activation of CARF-nucleases by cOA is pivotal for Type III immunity, this process can potentially lead to cell toxicity, thereby necessitating the requirement for regulatory mechanisms. Consequently, certain organisms possess CARF proteins that, with specific amino acid substitutions near the cOA binding site, function as either independent ring or self-limiting nucleases to remove cOA molecules (Athukoralage et al, 2019; Athukoralage et al, 2018; Du et al, 2023; Garcia-Doval et al, 2020; Jia et al, 2019b; Samolygo et al, 2020).

In contrast to the well-characterized CARF nucleases, much less is known about CARF-fused adenosine deaminases (Cad1), members of the CARF5 clade group. These CARF proteins possess the characteristics indicative of cOA binding and/or cleavage activities (Makarova et al, 2020), suggesting the possible cOA-regulated deamination functionality. Adenosine deaminases (ADA) are enzymes that convert adenosine and deoxyadenosine (A) to inosine and deoxyinosine (I), respectively (Carter, 1995). Infection-triggered A-to-I conversion could potentially deplete the pool of (deoxy)adenosine required for viral replication and thus virulence.

Two studies recently reported functional and structural characterization of Cad1 enzymes, *Lng*CAAD (Li et al, 2025) and *Bab*Cad1 (Baca et al, 2024), respectively. Instead of A-to-I conversion, both studies similarly discovered that Cad1 mediates the cOA-triggered ATP-to-ITP activities. In addition, Cad1 forms a hexameric assembly. Interestingly, the two studies differ in the mechanism of activation. Upon cOA binding, whereas a significant conformational change was observed in *Lng*CAAD (Li et al, 2025), a subtle sidechain adjustment of a single ADA catalytic histidine of *Bab*Cad1 was detected (Baca et al, 2024). Based on the conformational changes and the observed Hill coefficient of deamination, the study of *Lng*CAAD presents a model of cooperative regulation, however, with limited structural support (Li et al, 2025). In addition, the two studies disagree on the roles of Mg$^{2+}$, whereas Mg$^{2+}$ is believed to facilitate ATP binding to *Lng*CAAD (Li et al, 2025), it is believed to participate in catalysis in *Bab*Cad1 (Baca et al, 2024). Thus, the molecular mechanism underlying cOA-mediated activation remains unclear.

Here, we report biochemical and structural characterization of Cad1 from *Thermoanaerobaculum aquaticum* (*Taq*Cad1) that is distinct from the CARF proteins characterized to date. The CRISPR locus of the *T. aquaticum* genome encodes the known components for spacer integration, the Cas proteins constituting the Type III-B CRISPR-Cas effector complex, ancillary nuclease Csx1, and Cad1 (Fig. 1A). We show that Cad1 binds specifically to and acts as a ring nuclease for the cyclic tetra-adenosine (cA$_4$) in the regulation of the nuclease activity of Csx1, and the ADA domain of Cad1 converts ATP to ITP. Supported by biochemical data, the cryoEM structures of *Taq*Cad1 in the absence and presence of cA$_4$ and ATP reveal a

hexameric architecture and its interaction with the ligands. Structural classification and comparison across the three functional states of *Taq*Cad1 unveiled a dynamic protein network unique to Cad1 that mediates its activation by cA$_4$. We detected sequential conformational changes propagating through the hexamer, supporting the previously proposed cooperative activation mechanism.

## Results

### *Taq*Cad1 binds and hydrolyzes cA$_4$

The CARF domain of *Taq*Cad1 contains characteristic motifs for binding cOA molecules (Fig. 1A). Motif-II residues are crucial for cOA binding, while motif-I residues are largely responsible for cOA degradation. To confirm this activity and to distinguish the types of cOA molecules it interacts with, we employed microscale thermophoresis (MST) in measuring binding dissociation constants of the recombinantly expressed *Taq*Cad1 and its variants (Fig. 1B; Appendix Fig. S1). Titration of 0.1 nM–4 μM cA$_4$ to 50 nM fluorophore-labeled *Taq*Cad1 revealed an apparent disassociation constant of $13 \pm 7$ nM (Fig. 1B), similar to those measured for cA$_4$ binding to Sso2081 (Crn1) (Du et al, 2023) or Sis0811 (Molina et al, 2021) CARF proteins. In contrast, MST titration of cA$_6$ with *Taq*Cad1 resulted in an estimated apparent dissociation constant of ~1.4 μM (Appendix Fig. S1B), suggesting that cA$_4$ is the preferred ligand. Furthermore, mutation of the motif-II residue, Lys106, to alanine (K106A) largely abolished binding of cA$_4$ (Fig. 1B).

To evaluate if *Taq*Cad1 possesses the ring nuclease activity, we incubated it with cA$_4$ or cA$_6$ for 1 h at 37 °C and detected the reaction products by high-performance liquid chromatography (HPLC) and mass spectrometry. Whereas cA$_4$ was readily degraded (Fig. 1C), cA$_6$ remained intact (Appendix Fig. S1C). The products of degradation are predominantly di-adenosine cyclic phosphate (A$_2$ > P) and some adenosine cyclic phosphate (A > P) (Appendix Fig. S1D). Furthermore, the K106A variant did not degrade cA$_4$, consistent with its impaired binding to cA$_4$ (Fig. 1C). As shown for other ring nucleases (Athukoralage et al, 2018; Jia et al, 2019a; Niewoehner and Jinek, 2016), cA$_4$ degradation by *Taq*Cad1 did not require any metal ions, suggesting a general acid-base mechanism resembling that of RNase A (Yang, 2011). Finally, the ring nuclease activity of *Taq*Cad1 functions as a likely regulator of the co-expressed *Taq*Csx1 by inhibiting its cA$_4$-dependent ribonuclease activity, and this regulatory function depends on cA$_4$ concentration (Fig. 1D).

### *Taq*Cad1 lacks adenosine deamination activity

The high sequence homology of the ADA domain of *Taq*Cad1 to the known adenosine deaminases prompted us to investigate whether it promotes the conversion of A-to-I. For this, we made use

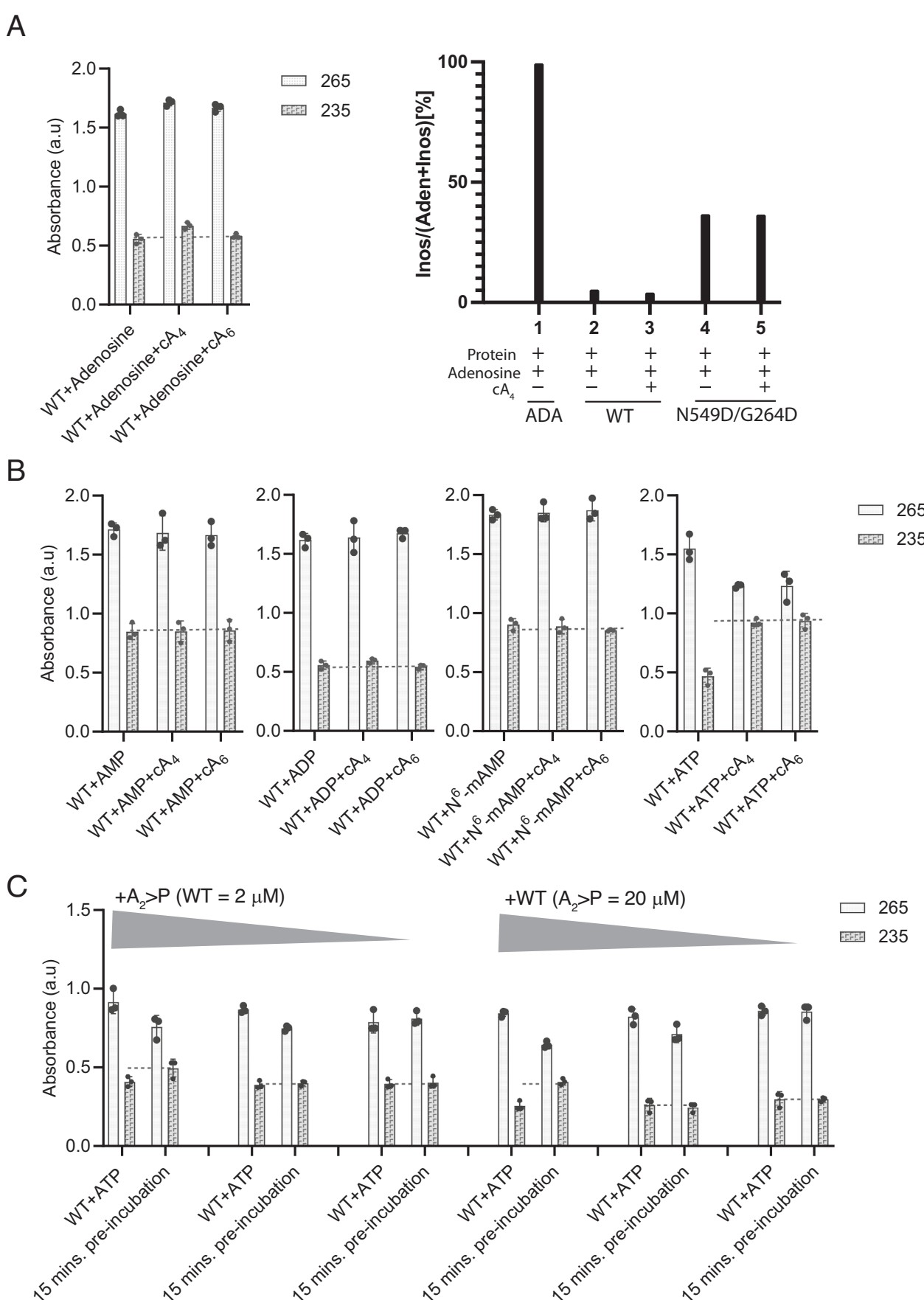

◀ **Figure 2.  UV absorbance and mass spectrometry analysis of TaqCad1 deamination activities.**

(A) Detection of spectroscopic or mass changes by using TaqCad1 wildtype (WT) with adenosine. Left, UV absorbance results. Right, mass spectroscopy results. "ADA" denotes the product of a control canonical adenosine deaminase. UV absorbance at wavelengths 265 and 235 nm measures the characteristic absorbance of adenosine and inosine, respectively. (B) Detection of spectroscopic changes by using TaqCad1 wildtype (WT) with AMP, ADP, $N^6$-methyl-AMP ($N^6$-mAMP), and ATP. UV absorbance at wavelengths 265 and 235 nm measures the characteristic absorbance of adenosine and inosine, respectively. TaqCad1-mediated reactions were dephosphorylated by calf intestinal alkaline phosphatase following heat deactivation. (C) Activation of ATP deamination by the ring nuclease product, primarily $A_2 > P$, of TaqCad1 at various $A_2 > P$ or TaqCad1 (WT) concentrations. Left, ring nuclease products resulted from incubating $cA_4$ with 2 μM WT TaqCad1 for 15 min, followed by heat inactivation, were used at 5, 10, and 20 μM, respectively, to activate ATP deamination. Right, TaqCad1 at 0.2, 0.5, and 2.0 μM were used to convert ATP in the presence of 20 μM pre-incubated $cA_4$ cleavage product. "WT + ATP" indicates the reaction in the absence of any ligand. TaqCad1-mediated reactions were dephosphorylated by calf intestinal alkaline phosphatase following heat deactivation. For panels (A–C), data represent the mean ± standard deviation of the mean ($n = 3$ biological replicates). Source data are available online for this figure.

of the characteristic change in UV absorption from adenosine at 265 nm to inosine at 235 nm. However, we were not able to detect any A-to-I activities, either in the presence or absence of $cA_4$ and $cA_6$ (Fig. 2A). We also employed mass spectrometry to detect the possible A-to-I conversion under similar conditions used for the spectroscopic method. We found that the wild-type TaqCad1 exhibits a weak A-to-I conversion (Fig. 2A).

Notably, the putative active sites of TaqCad1 and its close homologs contain two substitutions from the canonical ADA enzymes, a catalytic aspartate (Asp296 in mouse ADA) (Sideraki et al, 1996) by asparagine (Asn549 in TaqCad1) and a substrate-binding aspartate (Asp19 in mouse ADA) (Sideraki et al, 1996) by glycine (Gly264 in TaqCad1) (Fig. 1A). Previous studies showed that mutation of the Asp549-equivalent aspartate, Asp296, to asparagine in mouse ADA severely impaired substrate binding and catalytic efficiency (Sideraki et al, 1996). Furthermore, the carboxylate of Asp19 of mouse ADA forms two hydrogen bonds with the bound adenosine, one to its 5'-hydroxyl and another to its 3'-hydroxyl oxygen, respectively (Wilson et al, 1991). Substitution of both residues in TaqCad1 could thus render it catalytically inactive. However, replacing these residues with the canonical ADA residues (N549D/G264D) failed to rescue the low deamination activity of TaqCad1 (Fig. 2A), although this variant did increase the activity, indicating that these residues are not the sole or primary factors contributing to its lack of adenosine deamination activity.

## TaqCad1 converts ATP to ITP

We expanded our search for the substrate to other metabolites that contain an adenosine moiety by measuring their UV absorbance changes. We included AMP, ADP, ATP, $N^6$-methyl-AMP that are known substrates for related adenosine deaminases (Fig. 2B). Among these, ATP exhibited significantly strong conversion of inosine absorbance in the presence of $cA_4$ or $cA_6$ (Fig. 2B), suggesting the possibility that it is the substrate for TaqCad1, especially given the previous example of ATP to ITP conversion by the deaminase of the anti-phage system, RADAR (Duncan-Lowey et al, 2023; Gao et al, 2023) and the homologous Cad1 enzymes (Li et al, 2025; Baca et al, 2024).

We next assessed how the ring nuclease activity of TaqCad1 influences its deamination activity. We incubated $cA_4$ with the wild-type TaqCad1 for timed intervals (0–60 min) to ensure $cA_4$ degradation. The resulting cleavage products, primarily $A_2 > P$, were then tested for their ability to activate TaqCad1-mediated ATP-to-ITP conversion at varying enzyme concentrations. We found that while submicromolar TaqCad1 concentrations (<500 nM)

failed to initiate nucleotide conversion, full catalytic activity was observed at elevated concentrations (≥2 μM) (Figs. 2C and EV1). The same concentration-dependent activation was observed when $A_2 > P$ concentration was varied to higher than that of $cA_4$ capable of triggering deamination (Figs. 2C and EV1). This threshold effect suggests a reduced binding affinity of $A_2 > P$ for TaqCad1 compared to native $cA_4$ and that increasing either the TaqCad1 enzyme or $A_2 > P$ levels effectively rescued the impaired nucleotide conversion activity. Consistently, the same threshold effect on Csx1 activation in the presence of ring nuclease products of TaqCad1 was also observed (Fig. 1D).

We further confirmed the deamination of ATP by TaqCad1 by analyzing the reaction products using high-performance liquid chromatography (HPLC). Indeed, in the presence of either $cA_4$ or $cA_6$, as well as a divalent ion ($Mg^{2+}$ or $Ca^{2+}$), ATP was readily converted to ITP (Fig. 3A,B) while AMP, ADP, or $N^6$-methyl-AMP were not (Fig. EV2). The deamination activity requires $Mg^{2+}$ or $Ca^{2+}$, with $Ca^{2+}$ showing weaker efficacy (Fig. 3B). Interestingly, the N549D or the N549D/G264D variant of TaqCad1 did not react to ATP (Figs. 3C and EV2D), suggesting a distinct active site of TaqCad1 from the canonical adenosine deaminases. Furthermore, the ring-nuclease variant, K106A, also failed to convert ATP (Figs. 3C and EV2D), consistent with its lack of binding to $cA_4$ (Fig. 1B). Our data suggest that TaqCad1 has a different substrate requirement than canonical adenosine deaminases and that ligand binding to its CARF domain is critical for triggering its deamination activity.

## TaqCad1 forms a dynamic homohexameric assembly

To learn the structural basis, we obtained cryoEM structures of the wild-type TaqCad1 without a ligand or in the presence of $cA_4$ and ATP. 2D classifications reveal a clear hexameric architecture composed of three dimers, but also indicate substantial conformational heterogeneity, regardless of the presence of $cA_4$ (Figs. EV3–5; Appendix Tables S1, 2). To elucidate the extent of structural heterogeneity while achieving best possible resolution, we performed dual-structure reconstructions for each complex. Capitalizing on the inherent pseudo-C3 symmetry of the hexamer, we extracted all dimers by symmetry expansion while tracking their originating hexamers. Subsequent conformation-based classification of the dimers enabled high-resolution refinement, yielding optimized maps that resolve critical features of dimer architecture and ligand binding (Figs. EV3–5; Appendix Tables S1, 2). The classified dimers were then mapped back to the originating hexamers, revealing those with one, two, or three intact dimers, respectively. The hexamers with three intact dimers were modeled, leading to the best hexamer structures. For the $cA_4$- and

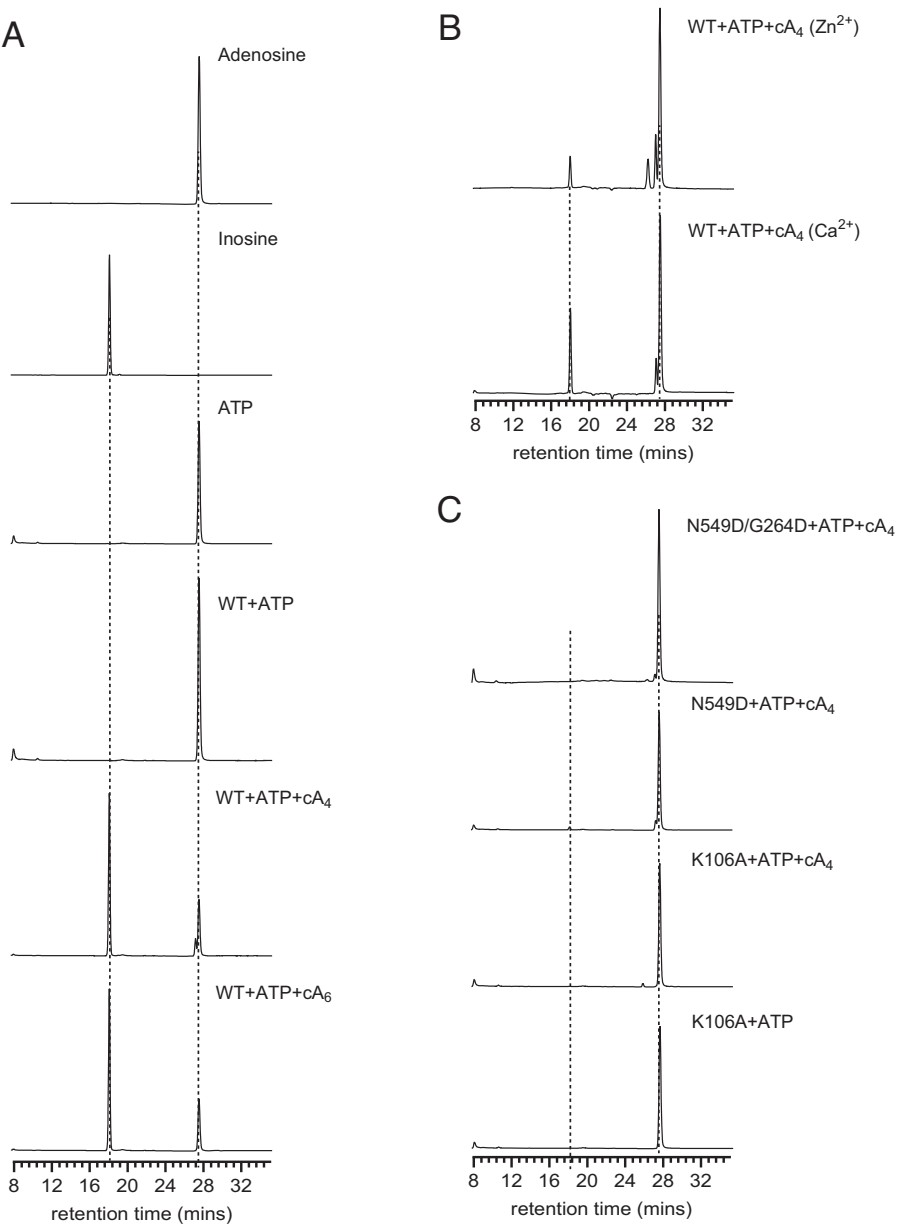

**Figure 3. High Performance Liquid Chromatography (HPLC) analysis of the possible products of ATP treated with *Taq*Cad1 or its variants.**

*Taq*Cad1-mediated reactions were dephosphorylated by calf intestinal alkaline phosphatase following heat deactivation. (A) HPLC elution profiles of ATP reacted with *Taq*Cad1 in the presence or absence of $cA_4$ or $cA_6$. (B) HPLC elution profiles of ATP reacted with *Taq*Cad1 in the presence of $cA_4$ and $Ca^{2+}$ or $Zn^{2+}$. (C) HPLC elution profiles of ATP reacted with *Taq*Cad1 variants in the presence of $cA_4$ and $Mg^{2+}$. Source data are available online for this figure.

ATP-bound *Taq*Cad1, the hexamers with one or two intact dimers were also reconstructed for comparing individual subunit structures within the hexamers. 3D variability analysis revealed motions of the dimers, even in the presence of $cA_4$ and ATP, revealing the dynamic nature of *Taq*Cad1 assembly that underlies the conformational heterogeneity (Movies EV1–3).

*Taq*Cad1 assembles from three pairs of monomers (CARF dimer) to form the hexamer with a pseudo-D3 symmetry (Fig. 4). The ADA domains create a donut-shaped core, while the CARF domains extend outward in a propeller-like configuration (Fig. 4).

A coordinated metal ion is observed at the conserved active site of the ADA domain (Fig. 5A), despite that no divalent ions were added during sample preparation. Given the similarity to the canonical zinc-containing ADA enzymes, we modeled the ion as $Zn^{2+}$. In the $cA_4$- and $cA_4$, ATP-bound complexes, two cleaved $A_2 > P$ molecules are observed in the CARF domain (Fig. 5A; Appendix Fig. S2). Finally, one ATP molecule and two accompanying $Mg^{2+}$ ions were observed in each of the ADA domains in the $cA_4$, ATP-bound complex (Fig. 5A), similar to what was reported for *Lng*CAAD (Li et al, 2025). Both the adenine ring and

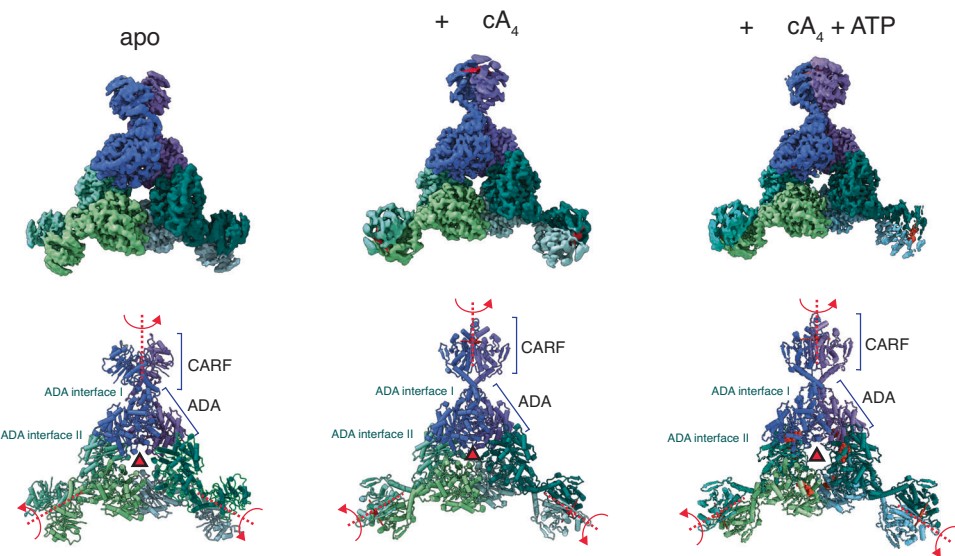

**Figure 4.  Structure overview of *Taq*Cad1 at three different functional states.**

Left, apo; Middle, with cA$_4$; Right, with cA$_4$ and ATP. Top, cryoEM density maps of the hexamers; Bottom, cartoon models of the hexamers. The pseudo-threefold axis is indicated by a filled solid triangle, while the three perpendicular twofold axes are indicated by dashed lines in red. Subunits are colored differently. The CARF and ADA domains are marked. "ADA interface I" and "ADA interface II" indicate the dimerization and the hexamerization interfaces via the ADA domains, respectively.

the triphosphate moiety of ATP are recognized by *Taq*Cad1 residues, including Asp549 and Gly264, that distinguish Cad1 from canonical ADAs (Figs. 1A and 5B). The observed interactions of these residues with ATP explain their essential role in ATP deamination (Figs. 2A, 3C and EV2D). Furthermore, the presence of Mg$^{2+}$ suggests its involvement in ATP binding, providing the molecular basis for the requirement of Mg$^{2+}$ in ATP deamination (Fig. 5B). Given that elevated A$_2$ > P levels maintain ATP deamination (Figs. 2C and EV1), the observed structures trapped with A$_2$ > P likely represent activated *Taq*Cad1.

Superimposition of the apo with the cA$_4$-bound structures revealed large differences in both the CARF and the ADA domains (RMSD 4.1 Å for 453 Cα atoms) (Fig. 5C), consistent with cA$_4$-mediated structural remodeling as previously reported for *Lng*CAAD (Li et al, 2025). Interestingly, a notable rotation of His461 that coordinates with the catalytic zinc is observed (Fig. 5C), similar to that observed in *Bab*Cad1 (Baca et al, 2024). We also observed an inward shift of another zinc coordination residue Asp548 and the ATP-binding residue Asn549 (Fig. 5C). Superimposition of the cA$_4$-bound with the cA$_4$, ATP-bound structures, on the other hand, revealed a strong overall (RMSD 0.9 Å for 583 Cα atoms) as well as the active site homology (Fig. 5C), further suggesting that cA$_4$ or A$_2$ > P binding pre-organizes *Taq*Cad1 for ATP binding.

## cA$_4$-mediated ADA remodeling activates TaqCad1

The availability of structures of three functional states allowed us to quantify cA$_4$-mediated changes. *Taq*Cad1 dimer is formed through both the CARF and ADA domains. The Rossmann fold core of the CARF domain (second P-loop and α6, residues 105–120) and the helix-turn-helix insertion (residues 211–229) comprise the CARF dimerization interface (Fig. 6A; Appendix Fig. S2). The ADA domain forms helical contacts involving α21- α23 (ADA interface I,

Fig. 6A; Appendix Fig. S2). Both the CARF and the ADA interfaces are significantly reshaped upon cA$_4$ binding. The CARF interface undergoes a significant increase in the buried surface area, from 1161 Å$^2$ in the apo to 1753 Å$^2$ in the cA$_4$-bound state (~50% gain). In contrast, ADA interface I exhibits an opposite change in buried surface area, decreasing from 1236 to 893 Å$^2$ (~38% loss). Evidence for protein dynamics was observed at the ADA interface I, where a conserved cation-π interaction between Trp472 of one ADA and Arg453 of another ADA displays two distinct conformations in the cA$_4$-bound but not in the apo state (Fig. 6A). The alternative conformations upon cA$_4$ binding can compensate for the entropic loss in structural tightening, thereby facilitating the rearrangement. Mutation of the equivalent tryptophan in *Lng*CAAD abolished its activity (Li et al, 2025). We designated the apo dimer as the relaxed (R) state and the cA$_4$-bound dimer as the taut (T) state (Fig. 5A).

To form the hexamer, each ADA domain further dimerizes through ADA interface II with another neighboring ADA (Figs. 4, 5A, and 6A). Upon cA$_4$ binding, ADA interface II also reshapes but in an opposite trend compared to ADA interface I. In the apo state, ADA interface II buries 863 Å$^2$ of surface area, which expands to 1029 Å$^2$ upon cA$_4$ binding, pushing the ADA domains closer together and a notable closure of the ADA hexameric ring.

Upon cA$_4$ binding, a near 70° twist of the CARF domains around the twofold axis propagates to the fused ADA domains. A well-conserved arginine residue, Arg397, on helix α21 at interface II swings towards the adjacent helix α21' in the apo state, disrupting this secondary structure in the cA$_4$-bound state (Fig. 6B). This "twist-and-squeeze" action by cA$_4$, facilitated by the inherent dynamics within the *Taq*Cad1 dimer, provides a potential entry path for ATP binding.

The importance of ADA interface II in *Taq*Cad1 activity is confirmed by mutational analysis. We mutated two residues on helix α21: Arg408 that forms an ion pair with neighboring α21

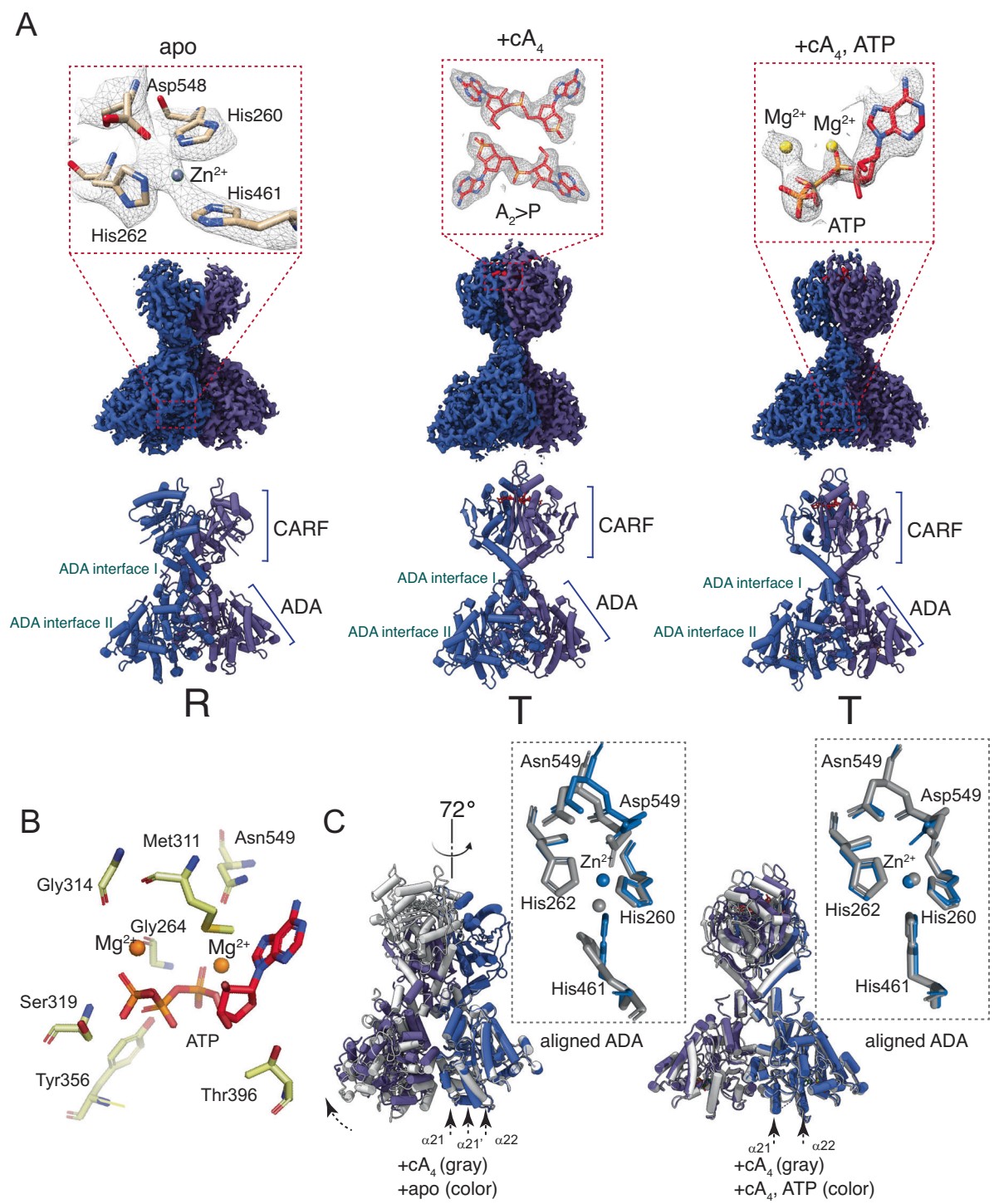

**Figure 5. Key features of *Taq*Cad1 dimers.**

(A) Density maps (top) and cartoon models (bottom) of the TaqCad1 dimers resulted from symmetry expansion at three different functional states. Left, apo; Middle, with cA₄; Right, with cA₄ and ATP. Insets show close-up views of density around the ADA active center (left), the bound $A_2 > P$ (middle), and the bound ATP (right). "ADA interface I" and "ADA interface II" indicate the dimerization and the hexamerization interfaces via the ADA domains, respectively. "R" denotes the Relaxed dimer. "T" denotes the Taut dimer. (B) Close contacts of a bound ATP with $Mg^{2+}$ and surrounding *Taq*Cad1 residues. (C) Superimposed structures between apo (gray) and the cA₄-bound (color) states (left) and those between the cA₄- (gray) and cA₄, ATP-bound (color) states (right). Insets compare ADA catalytic residues between the apo (gray) and the cA₄-bound (blue) states (left) and between the cA₄- (gray) and cA₄, ATP-bound (blue) states (right). The aligned ADA domain is indicated. Arrows indicate the rotation from the cA₄-bound to the apo state. Remodeled secondary structures are indicated.

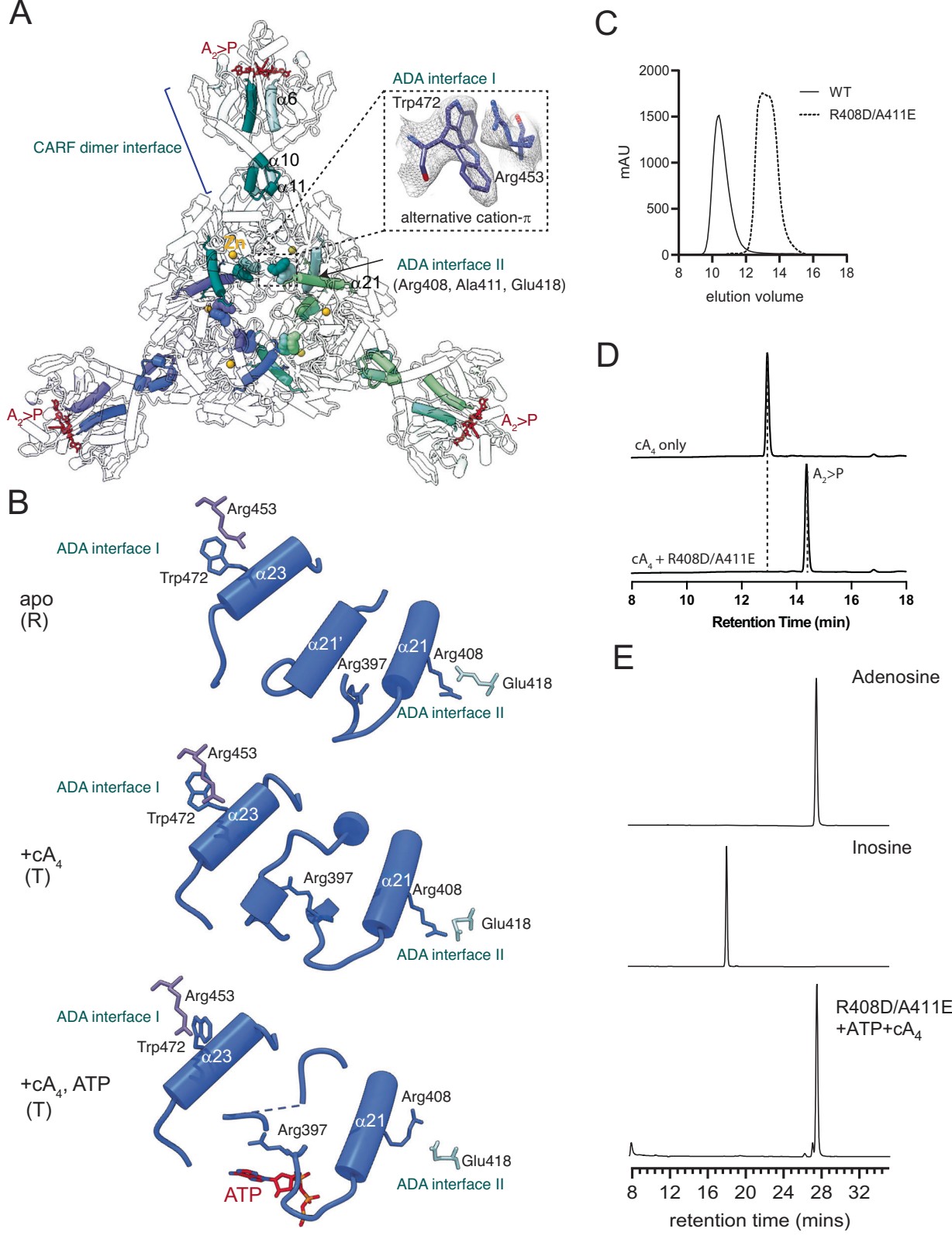

**Figure 6. Identification of a protein network facilitating cA$_4$-mediated activation.**

(A) Highlight of the dimerization and hexamerization interfaces mediated by the CARF and ADA domains, respectively. Helices α6, α10, and α11 mediate dimerization while α21 mediates hexamerization. These helices are shown as solid cylinders and labeled. The bound A$_2$>P molecules are shown as red stick models. "ADA interface I" and "ADA interface II" indicate the dimerization and the hexamerization interfaces via the ADA domains, respectively. Key residues of both interfaces are labeled. (B) Structure remodel near the ATP entry path cA$_4$ binding. The key helices under transition, flanked by ADA interfaces I & II, are shown as cartoons and key residues as stick models. The corresponding conformation of each structure is labeled by R (relaxed) or T (taut), respectively. (C) Gel filtration elution profiles of the wild-type (WT) and R408D/A411E variant (R408D/A411E) indicate significantly reduced hydrodynamic radius upon mutation. (D) HPLC profiles of the digested cA$_4$ molecule by the R408D/A411E mutant (bottom) in comparison with cA$_4$ alone (top). (E) HPLC analysis of ATP deamination by the wild-type (WT) and the R408D/A411E mutant in the presence of cA$_4$.

Glu418 and Ala411 that is buried at the α21-α21 interface to aspartate and glutamate, respectively (R408D/A411E) (Fig. 6A). We showed by the altered size-exclusion elution profile that the R408D/A411E mutations disrupted the hexameric assembly (Fig. 6C). While the disassembled dimer remains active in cleaving cA$_4$ (Fig. 6D), it can no longer convert ATP to ITP (Fig. 6E), consistent with its critical role in the "twist-and-squeeze" activation model.

## Cooperative activation of ADA domains by cA$_4$

We next addressed if the cA$_4$-mediated changes within the dimer can propagate to adjacent dimers that are not bound with cA$_4$. We first analyzed ATP binding features in the heterogeneous cA$_4$, ATP-bound structures. We displayed the densities around the ATP binding sites for the hexamers containing one, two, or three intact CARF dimers (Fig. 7A). The hexamer containing a single CARF dimer shows weak but convincing densities for four bound ATP molecules, two linked to the intact CARF dimer and two immediately adjacent to them (Fig. 7A). The hexamer containing two intact CARF dimers shows clear densities for all six bound ATP (Fig. 7A), suggesting that the cA$_4$-mediated activation can be propagated.

We then analyzed the structural features of each ADA domain within the same set of cA$_4$, ATP-bound hexamers. Whereas the cA$_4$-bound and intact dimers are in the T conformation, the adjacent cA$_4$-free dimers adopt a hybrid R and T conformation (R/T) (Fig. 7B) including the notable alternative Trp472 conformation as observed in cA$_4$-bound dimers (Fig. 6A). Importantly, the active sites of the R/T ADA domains in the cA$_4$-free dimers display the similar sidechain rotation of His461 (Fig. 7B), consistent with the observed densities for ATP in these dimers. In some of the R/T ADA domains, there is also a corresponding disruption of α21' despite their CARF domains do not have a bound ligand. This observation suggests that binding of cA$_4$ to a CARF interface not only activates its linked but also the adjacent ADA domains, likely through the protein network formed by the ADA interfaces I & II (Fig. 6), providing the molecular basis for a sequential conformational activation model.

## TaqCad1 has a mild defense activity at 37 °C

Most CARF domain-containing proteins are involved in some defense function, and their activation often leads to either cell dormancy or abortive infection (Steens et al, 2022). Here, we investigated whether the enzymatic activity of *Taq*Cad1 on ATP causes decreased cell viability. Therefore, we co-expressed *Taq*Cad1 and a cA$_4$-producing type III complex from *Treponema succinifaciens* in *E. coli* BL21-AI. As controls, we also co-expressed a cA$_4$-responsive *T. succinifaciens* Card1 (*Tsu*Card1) and an empty vector with the *T. succinifaciens* type III complex as positive and negative controls, respectively (Rostol et al, 2021) (Appendix Fig. S3).

Target and non-target plasmids were used to express IPTG-inducible protospacers that were either complementary or non-complementary to the spacer, respectively (Ichikawa et al, 2017). These were electroporated into the abovementioned strains, and tenfold serial dilutions were plated onto IPTG-containing plates to induce the production of target or non-target RNA. When *Taq*Cad1 is present, the transformation efficiency of the target plasmid was significantly lower by a modest one order of magnitude compared to the non-target plasmid (Appendix Fig. S3). Interestingly, the presence of *Taq*Cad1 and target RNA led to visibly smaller colonies as compared to the non-target condition, suggesting slower growth rates (Appendix Fig. S3).

These suggest that *Taq*Cad1 has a minimal effect on growth. However, it should be noted that the assays were done at 37 °C, well below the expected optimal temperature of the protein (60 °C). This, in addition to the ring nuclease activity of *Taq*Cad1, may have minimized the cA$_4$-dependent defense activity of the protein. Altogether, these suggest that *Taq*Cad1 activation via cA$_4$ production upon target RNA detection has a mild defense response at 37 °C.

## Discussion

We provide evidence that an adenosine deaminase-like enzyme fused with a ring nuclease CARF apparently has the ability to convert ATP to ITP in a cOA-dependent manner. In addition, the ring nuclease of *Taq*Cad1 regulates the RNA shredding activity of another co-functional CARF nuclease, Csx1. Our structural analysis shows that the CARF domain forms the canonical cA$_4$-binding and degradation module, whereas the ADA-like domain mediates oligomerization and interacts with ATP. Thus, *Taq*Cad1 is a dual functional enzyme mediated by cOAs, both as a ring nuclease in regulating Csx1 and itself, and as a deaminase to accumulate the level of ITP. Despite the robust enzymatic activities in vitro, interestingly, an in vivo plasmid challenge assay revealed a mild defense activity.

Previously, a study identified that the anti-phage system, RADAR, possesses an adenosine deaminase subunit, RdrB, that also converts ATP or dATP to ITP or dITP, respectively (Duncan-Lowey et al, 2023; Gao et al, 2023). It was suggested that the accumulation of ITP or dITP achieves anti-phage activity by both influencing the nucleotide pool and/or nucleic acid synthesis (Duncan-Lowey et al, 2023). TaqCad1 may exert the same cellular effects upon activation by cOAs, although its efficient ring nuclease

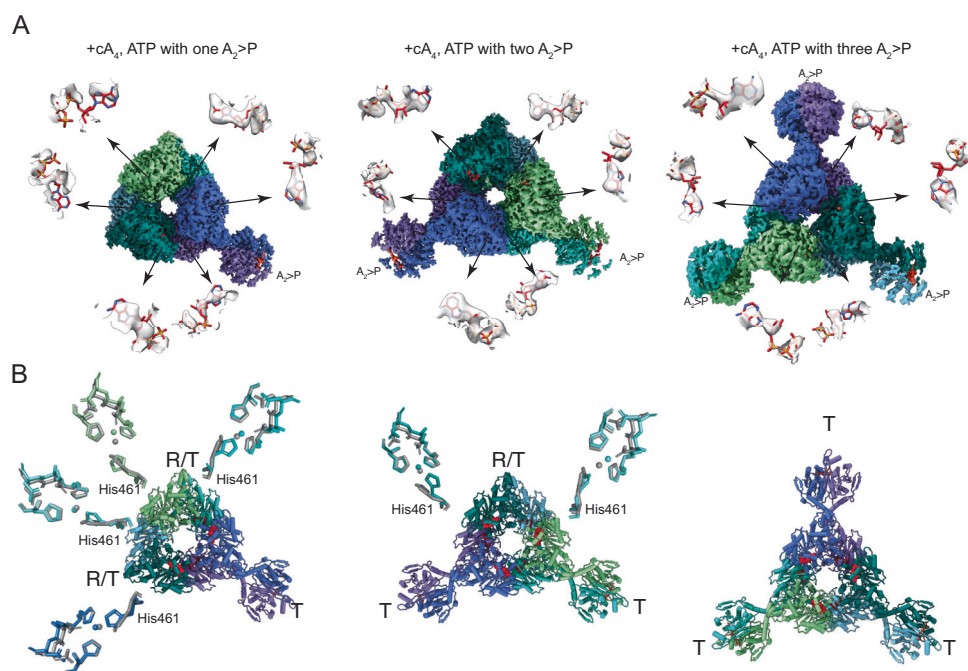

**Figure 7.  Sequential activation of ADA domains of *Taq*Cad1 upon cA$_4$ binding.**

(A) Overall density for the cA$_4$, ATP-bound hexamers containing one (left), two (middle), and three (right) intact dimers, respectively. Densities around each of the six indicated ATP binding sites within the three types of hexamers are shown at an equal threshold. (B) Cartoon models corresponding to the density shown in panel A for the cA$_4$, ATP-bound hexamers containing one (left), two (middle), and three (right) intact dimers, respectively. The observed conformation at each time is labeled as T (taut), R (relaxes), or the hybrid between R and T (R/T), respectively. The catalytic residues for each ADA domain in R/T form are shown as stick models along with those of the apo state (gray). Zinc ions are shown as small spheres. The modeled ATP molecules are shown as red spheres.

activity may prevent a long-lasting effect from being detected at a significant level. Interestingly, whereas RADAR is able to deaminate a number of adenosine-containing metabolites, *Taq*-Cad1 is specific for ATP, which highlights their structural differences (Appendix Fig. S4).

While CARF-proteins primarily exist as homodimers (Jia et al, 2019b; Niewoehner and Jinek, 2016; Rostol et al, 2021; Zhu et al, 2021), some instances show their presence as a monomer (McMahon et al, 2020) and hexamer (Molina et al, 2019). Regardless of the oligomeric configuration, these proteins detect cOA activations through the linkage from the CARF to the effector domain. *Taq*Cad1 forms a hexamer mediated by the ADA domain and is capable of binding three cA$_4$ or the weaker ligand cA$_6$. The fact that ATP-to-ITP conversion is only observed upon cA$_4$ or cA$_6$ binding suggests an allosteric regulation of ATP binding by the second messenger produced upon viral infection. We identified a network of interactions within *Taq*Cad1, supported by oligomerization uniquely evolved in *Taq*Cad1, that propagates structural changes from the CARF domain to the ATP binding sites via a twist-and-squeeze action. Furthermore, the activation process is cooperative through sequential R-to-T conformational conversions.

The large conformational changes in *Taq*Cad1 upon cA$_4$ binding resemble those observed in *Lng*CAAD (Li et al, 2025). Interestingly, the more subtle but convincing sidechain rotation of the catalytic His461 was observed in *Bab*Cad1 that, interestingly, did not exhibit gross conformational changes (Baca et al, 2024). We propose that this observation underscores the structural and

mechanistic diversity of CARF enzymes, which employ distinct conformational dynamics to enable activation. Notably, certain conformational changes—such as those in *Bab*Cad1—appear undetectable with the current methods. This raises intriguing questions: Are these CARF proteins capable of cooperative regulation? If not, could this limitation arise from their unique cellular context rather than intrinsic molecular constraints?

Our study, as well as those of *Lng*CAAD and *Bab*Cad1, demonstrated that Cad1 enzymes exhibit a strong preference for one cOA ligand (cA$_6$ for *Bab*Cad1, cA$_4$ for *Lng*CAAD and *Taq*Cad1), but can also be activated by another cOA with considerably weaker binding affinity (cA$_4$ for *Bab*Cad1, cA$_6$ for *Lng*CAAD and *Taq*Cad1). Our observed concentration threshold effect further adds the ring nuclease cleavage product, A$_2$ > P, as an activator to *Taq*Cad1, if present at high concentrations. Notably, all three studies captured the structures of the enzymes bound with a low-affinity ligand and ATP, raising the question of whether they truly represent activated structures. We showed that, in the presence of A$_2$ > P, *Taq*Cad1 ADA domains are sequentially modulated and bound by ATP, supporting that the captured structures do represent the true activation process. Furthermore, comparison of the structures, particularly between *Taq*Cad1 and the homologous *Lng*CAAD, revealed similar ATP-bound conformations irrespective of the type of weak-affinity ligand (Appendix Fig. S5), further validating their active status. Analysis of the recently available structures highlights the unique value presented in this study that significantly expands the molecular basis for cooperative activation.

# Methods

### Reagents and tools table

| Reagent/resource | Reference or source | Identifier or catalog number |
|---|---|---|
| **Experimental models** | | |
| *E. coli* DH5a | ATCC | 67879 |
| *E. coli* NiCo(DE3) | New England BioLabs | C2529H |
| *E. coli* TOP10 | Invitrogen | C404010 |
| **Recombinant DNA** | | |
| pET28a(+) | Florida State University Molecular Cloning Facility | N/A |
| **Oligonucleotides and other sequence-based reagents** | | |
| PCR primers | This study | Appendix Table S3 |
| **Chemicals, enzymes and other reagents** | | |
| LB Broth, Miller | VWR | J106-2KG |
| Kanamycin Monosulfate, USP Grade | GoldBio | 25389-94-0 |
| Agar, bacteriological | VWR | J637-1KG |
| Agarose | Fisher BioReagents | BP160-500 |
| Isopropyl-β-D thiogalactopyranoside (IPTG) | Sigma-Aldrich | 367-93-1 |
| Sodium chloride | Sigma-Aldrich | S7653 |
| (4-(2-hydroxyethyl)-1-piperazineethanesulfonic acid) (HEPES) sodium salt | Sigma-Aldrich | H7006-500G |
| Imidazole | ACROS Organics | 288-32-4 |
| Glycerol | Fisher Chemical | G33-4 |
| 2-Mercapotethanol | VWR | M131-250mL |
| Magnesium chloride | Sigma-Aldrich | M8266-100G |
| Calcium chloride | Sigma-Aldrich | C5670-100G |
| Nickel (II) sulfate hexahydrate | Sigma-Aldrich | 227676-100 G |
| Sodium dodecyl sulfate | VWR | 0227-100 G |
| Gel loading dye, Purple | New England Biolabs | B7025S |
| Q5 Site-directed Mutagenesis Kit | New England Biolabs | E0554S |
| E.Z.N.A. Plasmid DNA Miniprep Kit | Omega BIO-TEK | D6942-01 |
| Adenosine | Sigma-Aldrich | A9251-1G |
| Inosine | Sigma-Aldrich | I4125-1G |
| Adenosine 5′-monophosphate | Fisher Scientific | AC102790050 |
| Adenosine 5′-diphosphate sodium salt | Sigma-Aldrich | A2754-100MG |
| $N^6$-Methyladenosine 5′-monophosphate sodium salt | Fisher Scientific | 50-490-880 |
| dATP solution | New England Biolabs | N0440S |
| Adenosine 5′-Triphosphate (ATP) | New England Biolabs | P0756S |
| Quick CIP | New England Biolabs | M0525S |
| rCutSmart Buffer | New England Biolabs | B6004S |

| Reagent/resource | Reference or source | Identifier or catalog number |
|---|---|---|
| Urea | Fisher Biotech | BP169-212 |
| Acrylamide/Bis-acrylamide, 29:1 | Sigma-Aldrich | A2792-100ML |
| Ammonium Peroxydisulfate | Fisher Chemical | A682-500 |
| Tetramethylethylenediamine (TEMED) | OmniPur | 110-18-9 |
| Bromophenol Blue | Bio-Rad | 161-040 |
| PageRuler Prestained Protein Ladder | Thermo Fisher Scientific | 26616 |
| KLD Enzyme Mix 25 rxns | New England Biolabs | M0554S |
| SYBR Gold Nucleic Acid Gel Stain | Invitrogen | S11494 |
| **Software** | | |
| Graphpad Prism | https://www.graphpad.com/ | N/A |
| CryoSPARC | https://cryosparc.com/ | |
| Phenix | https://phenix-online.org/ | |
| Chimera | https://www.rbvi.ucsf.edu/chimerax/ | |
| Relion | https://github.com/3dem/relion | |
| Coot | https://www2.mrc-lmb.cam.ac.uk/personal/pemsley/coot/ | |
| Topaz | https://github.com/tbepler/topaz | |

## Cloning, protein expression, and purification

The complete *Taq*Cad1 sequence (1–625) was cloned into a pET28a vector, using NcoI and BamHI restriction sites, giving rise to *Taq*Cad1 fused with a C-terminal hexa-histidine tag. Mutations of the *Taq*Cad1 were introduced via Q5 mutagenesis (New England Biolabs) and confirmed by full plasmid sequencing (Eurofins Genomics). For overexpression, the protein-coding plasmids were transformed into NiCo21 (DE3) cells (New England Biolabs) that were then cultured at 37 °C to an OD$_{600}$ of 0.6 and induced with 0.3 mM isopropyl β-D-1-thiogalactopyranoside (IPTG) at 18 °C overnight. Cells were harvested by centrifugation and stored at −80 °C. Defrosted cell pellets were lysed via sonication in a lysis buffer comprising 20 mM HEPES at pH 7.5, 200 mM NaCl, 50 mM imidazole, 5% glycerol, and 5 mM phenylmethylsulfonyl fluoride (PMSF). After centrifugation at 16,000 RPM for 30 min, the supernatant was applied onto Nickel-NTA Resin (GE Healthcare), followed by extensive washing with the lysis buffer with 50 mM imidazole. The target protein was eluted using the lysis buffer with 500 mM imidazole. The elutant was loaded onto the Mono Q® 5/50 GL (Cytiva) column and eluted with a linear gradient from 100 mM to 1 M NaCl over 45 min. The peak fractions were collected and concentrated before being loaded onto the Superdex 200 16/60 column pre-equilibrated in the gel filtration buffer (20 mM Hepes,

pH 7.5, 150 mM NaCl, 5 mM ßME). The target proteins were collected and flash-frozen in liquid nitrogen for storage at −80 °C.

The complete *Taq*Csx1 sequence (1–458) was amplified from the *Thermoanaerobaculum aquaticum* genomic DNA and cloned into the pACYduet vector with a C-terminal hexa-histidine tag. *Taq*Csx1 was expressed in E. *coli* NICO cells and purified similarly to *Taq*Cad1 described above.

## Cryo-EM sample preparation, data collection, and 3D reconstruction

The apo, $cA_4$-bound, and $cA_4$-, ATP-bound *Taq*Cad1 complexes were both prepared with a final *Taq*Cad1 concentration at approximately 3 mg/mL (21 μM), stored in gel filtration buffer. For the $cA_4$-bound complex, 105 μM $cA_4$ was mixed and incubated with *Taq*Cad1 for 10 min at room temperature prior to grid making. For the $cA_4$-, ATP-bound complex, 200 μM $cA_4$, 1 mM ATP and 1 mM $MgCl_2$ were mixed and incubated with *Taq*Cad1 for 10 min at room temperature prior to grid making. About 4 μL of each complex was applied onto UltraAuFoil 300 mesh R1.2/1.3 grids that were glow-discharged with Solarus 950 plasma cleaner (Gantan). Following 30-second incubation inside the chamber of FEI Vitrobot Mark IV under 100% humidity at room temperature, the grids were blotted for 2.5 or 3 s and plugged into liquid ethane. The prepared grids were then stored in liquid nitrogen until data collection.

The cryoEM data of each complex were collected at the Laboratory for BioMolecular Structure of the Brookhaven National Laboratory or at Van Andel Institute cryoEM core using 300 kV Titan Krios (Thermo Fisher) equipped with an energy filter and K3 direct electron detector (Gatan) (Appendix Table S1). For the apo and $cA_4$-bound complexes, the movies were recorded at a nominal magnification of 81,000 in super-resolution mode, with energy filter slit set to 15 eV, corresponding to a corrected physical pixel size of 1.06 Å/pixel. An accumulated dose of 60 $e^-/Å^2$ was spread over 60 frames with random defocus set to −0.8 to −2.5 μm (Appendix Table S1). For the cA4-ATP-bound complex, the movies were recorded at a nominal magnification of 105,000 in super-resolution mode, with energy filter slit set to 20 eV, corresponding to a corrected physical pixel size of 0.83 Å/pixel (Appendix Table S1). An accumulated dose of 50 $e^-/Å^2$ was spread over 50 frames with random defocus set to −1.0 to −2.2 μm. Raw multi-frame micrographs were motion corrected by MotionCorr2 (Zheng et al, 2017) with a binning factor of 2. Following contrast transfer function (CTF) estimation by Gctf (Zhang, 2016), 4186, 8597, and 7862 high-quality micrographs for the apo, $cA_4$-bound, and $cA_4$-, ATP-bound complex, respectively, were selected for further processing.

Image processing was performed using CryoSPARC (Punjani et al, 2017) in combination with Relion (Zivanov et al, 2022). Briefly, particles were picked using either template picker or Topaz (Bepler et al, 2019), and subject to several rounds of 2D-Classification in CryoSPARC to sort out low-quality ones and contaminants. A total of 1,346,130, 4,121,609, and 606,589 particles for apo, $cA_4$-bound, and $cA_4$-, ATP-bound complex, respectively, were selected and exported for 3D-auto refine in Relion with C3 symmetry. The resulting particle star files were then symmetry expanded using relion_sym_expand command with the C3 operator. A mask around one of the Cad1 dimers was constructed and then used to extract masked densities while subtracting unmasked signals (Figs. EV3–5). The resulting dimer particles were subject to multiple rounds of alignment-free 3D classification in Relion to identify dimer particles with an intact CARF domain. Intact dimer particles were reconstructed and refined in CryoSPARC with Non-uniform refinement. To reconstruct the hexamer structures, the selected intact dimer particles were mapped back to identify hexamer particles with one, two, and three intact dimers by a custom Python script. For both the apo and $cA_4$-bound complexes, the hexamer containing three intact dimers were reconstructed and modeled. For the $cA_4$-, ATP-bound complex, the hexamers containing one, two, and three intact dimers were reconstructed and modeled (Appendix Table S2).

Structural models were built with manual adjustment in COOT (Emsley et al, 2010) and refined in PHENIX (Liebschner et al, 2019) to satisfactory stereochemistry and real-space map correlation parameters.

## UV absorbance analysis

*Taq*Cad1-mediated reactions were assembled by incubating 2 μM *Taq*Cad1 or its variants with or without 20 μM $cA_4$ or $cA_6$ (BIOLOG Life Science Institute) in a buffer containing 20 mM Tris-HCl, pH 7.5, 150 mM NaCl, and 1 μM $Zn^{2+}$, 1 mM $MgCl_2^{2+}$ or 1 mM $CaCl_2^{2+}$ and 20 μM various deamination ligands. The reactions were carried out at 55 °C for 30 min before being heated at 95 °C for 10 min. Calf intestinal alkaline phosphatase was added to dephosphorylate the products prior to urea treatment or following heat deactivation. Reactions were spun down at 13,2000 RPM for 20 min. UV absorbance of the reaction products was recorded on a NanoDrop spectrophotometer.

For the $A_2 > P$ activation assay, the same reaction mixtures described above were set up, containing either the WT or K106A variant of *Taq*Cad1 without ATP. Following incubation at 55 °C for 15, 30, and 60 min, respectively, the reaction mixtures were heat inactivated and cleared by centrifugation and added fresh 2 μM WT *Taq*Cad1 and 20 μM ATP. Calf intestinal alkaline phosphatase was added to dephosphorylate the products prior to urea treatment or following heat deactivation. UV absorbance of the reaction products was recorded on a NanoDrop spectrophotometer.

## HPLC analysis

High-performance liquid chromatography (HPLC) was employed to analyze ring nuclease activity and possible deamination products. For the ring nuclease activity assay, reactions were initiated by incubating 2 μM *Taq*Cad1 enzyme or its variants with 20 μM synthetic $cA_4$ or $cA_6$ (BIOLOG Life Science Institute) in a buffer containing 20 mM Tris-HCl, pH 7.5, 150 mM NaCl, and 1 μM $Zn^{2+}$. The reactions were carried out at 37 °C for 10–60 min before being heated at 95 °C for 10 min and spun down at 13,2000 RPM for 20 min.

For possible deamination activity, reactions were initiated by incubating 50 μM wildtype or mutant TaqCad1 enzymes, 1 mM ATP, and/or 100 μM synthetic $cA_4$ or $cA_6$ in a 100 μl reaction volume containing 20 mM Tris-HCl pH 7.5, 150 mM NaCl, 1 mM $MgCl_2$, 1 mM $CaCl_2$, or 10 μM $ZnCl_2$ at 55 °C for 30 min. The reaction mixtures were quenched by adding an equal volume of 8 M urea and cleared by centrifugation at 10,000 rpm for 10 min at 4 °C. Calf intestinal alkaline phosphatase was added to dephosphorylate the products prior to urea treatment or following heat deactivation.

The supernatant of both reactions was analyzed using an HPLC system (Shimadzu Prominence LC-20) fitted with a SunFire C18

column (4.6 mm × 150 mm, 3.5-μm particle size). A 5 μl sample volume was injected and separated by a linear gradient of eluent A (20 mM ammonium bicarbonate) and eluent B (100% acetonitrile) at a flow rate of 0.3 ml/min over a total 22-min duration. The gradient conditions were as follows: a gradient of 2–30% B from 0 to 12 min, from a transition to 95% B for washing from 12.1 to 16 min, a constant 95% B concentration from 16.1 to 17.0 min, and finally, return to 2% B for equilibration from 17.1 to 22.0 min.

## Mass spectrometry analysis

For mass spectrometry analysis, reaction mixtures were prepared identically to those used for HPLC. Following urea denaturing, the supernatant was analyzed on an Agilent 6230 TOF-MS with the Agilent Mass Hunter Workstation Software TOF 6500 series in positive ion mode. Spectrum was analyzed using Agilent Mass Hunter Qualitative Analysis Navigator v.B.08 and visualized using GraphPad Prism. The relative abundance of the $cA_4$ degradation products was calculated based on the area under the curve (AUC) of each specific ion in the extracted ion chromatograms, normalized on the AUC of adenosine.

## Microscale thermophoresis

The binding affinities of *Taq*Cad1 and its variants with cOA were evaluated by microscale thermophoresis (MST) binding assay, using the instrument of NanoTemper Monolith NT.115 (NanoTemper Technologies, Munchen, Germany). To track the movement of *Taq*Cad1 during the experiments, the protein was labeled using the His-tag labeling kit RED-tris-NTA (MO-L008, Nanotemper Technologies). The labeling was performed in PBST buffer, which also served as the assay buffer for MST experiments. After labeling, 50 nM *Taq*Cad1 protein was incubated on ice for 60 min with varying concentrations of cOA and then loaded into Monolith NT.115 MST standard-treated capillaries (MO-K022, NanoTemper Technologies). The measurements were carried out at 25 °C using a Monolith NT.115 instrument with MO. control software, setting the light-emitting diode (LED) or excitation power at 40–80%, and MST medium power. Data analysis was performed using GraphPad Prism.

## In vitro RNA cleavage assay

The RNA cleavage assays were performed in a cleavage buffer containing 20 mM MES, pH 6.5, and 5 mM $MnCl_2$. The reactions were performed at 60 °C for 20 min and contained 100 nM Csx1 and 1 μM target RNA (Appendix Table S3) and varying concentrations of *Taq*Cad1. The reactions were quenched using 2x formamide dye (95% formamide, 0.025% SDS, 0.025% bromophenol blue, 0.025% xylene cyanol FF, 0.5 mM EDTA). The reaction products were heated at 95 °C/3 min and separated by 7 M Urea, 15% polyacrylamide gel electrophoresis (PAGE) gels in 1x Tris/Borate/EDTA (TBE) running buffer and were visualized by staining with SYBR Gold II (Invitrogen) stain.

## In vivo plasmid challenge assays

To assess the effect of in vivo expression and activation of *Taq*Cad1, electrocompetent *E. coli* BL21-AI were co-transformed with two plasmids prior to target/non-target plasmid challenge.

The first one, pTsuTypeIII, carried several genes from *T. succinifaciens* for the formation of type III CRISPR-Cas complexes: *csm1* to *csm5*, *cas6*, and a minimal CRISPR array carrying one spacer sequence. These were individually placed under the control of T7 promoters (Appendix Fig. S3). The second plasmid, called a pEffector, encoded for one of the effectors used in the assay (either *Taq*Cad1 or *Tsu*Card1), constitutively expressed by a lacUV5 promoter (Appendix Fig. S3). On the target and non-target plasmids, a non-coding RNA was placed under the control of a trc promoter and lacO (Appendix Fig. S3) (Ichikawa et al, 2017). In the case of the target plasmid, this non-coding RNA is complementary to the spacer on pTsuTypeIII.

The transformations were carried out in triplicate, where 100 ng of either target or non-target plasmid were electroporated into each strain. After electroporation, the cells were recovered in SOC medium for 1 h at 37 °C. For calculating the relative transformation efficiencies, tenfold serial dilutions of the recovered cells were plated onto LB agar containing 34 μg/mL chloramphenicol, 50 μg/mL carbenicillin, 50 μg/mL kanamycin, 0.2% arabinose, and 0.5 mM IPTG. The plates were incubated overnight at 37 °C, and transformation efficiencies were quantified. Data were analyzed and visualized using R 4.3.0, with statistical significance calculated by a one-sided unpaired Welch's *t*-test.

For the liquid culture growth assay, the procedure for electroporation and recovery were done as previously described. After recovery, the cells were washed three times with LB medium. The cells were then diluted to $OD_{600} = 0.05$, and 150 μL of each dilution was added to a 96-well plate and covered with 50 μL mineral oil. The plate was incubated in a BioTek Epoch 2 Microplate Spectrophotometer for 22 h at 37 °C, with double-orbital shaking. Growth, measured by $OD_{600}$, was recorded every 10 min. Data were visualized using Python 3.10.

## Data availability

The atomic coordinates of the cryo-EM structures of the symmetry-expanded dimers for Apo *Taq*Cad1, *Taq*Cad1-$cA_4$, and *Taq*Cad1-$cA_4$-ATP have been deposited in the Protein Data Bank (https://www.rcsb.org) under the identifiers 9OF1, 9EBT, 9OFB, respectively. The corresponding maps have been deposited in the Electron Microscopy Data Bank (https://www.ebi.ac.uk/emdb/) under the entries 70419 (consensus), 70423 (CARF focused), 70424 (ADA focused) and 70417 (composite) for Apo *Taq*Cad1 dimer, 47886 (consensus), 47887 (CARF focused), 47888 (ADA focused), and 47890 (composite) for the *Taq*Cad1-$cA_4$, 70425 (consensus), 70426 (CARF focused), 70427 (ADA focused), and 70422 (composite) for *Taq*Cad1-$cA_4$-ATP. The atomic coordinates of the cryoEM structures of the Apo *Taq*Cad1, *Taq*Cad1-$cA_4$, and T*aq*Cad1-$cA_4$-ATP hexamers with three intact dimers have been deposited in the Protein Data Bank (https://www.rcsb.org) under the identifiers 9EKA, 9MMW, 9OFC, and maps in the Electron Microscopy Data Bank (https://www.ebi.ac.uk/emdb/) under the entries 48116, 48405, and 70428, respectively. Those for the TaqCad1-$cA_4$-ATP hexamers with one and two intact dimers have been deposited in the Protein Data Bank (https://www.rcsb.org) under the identifiers 9OFD and 9OFE and maps in the Electron Microscopy Data Bank (https://www.ebi.ac.uk/emdb/) under the entries 70429 and 70430. respectively.

The source data of this paper are collected in the following database record: biostudies:S-SCDT-10_1038-S44318-025-00578-y.

## Peer review information

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

## Acknowledgements

Authors acknowledge the use of instruments at the Biological Science Imaging Resource (BSIR) supported by Florida State University, the Laboratory for BioMolecular Structure (LBMS), the Pacific Northwest Center for Cryo-EM (PNCC), and the David Van Andel Advanced CryoEM Suite (RRID:SCR_023210) at the Van Andel Institute Authors thank X. Fu of BSIR for assistance in cryoEM data collection; G. Seo of the Institute of Molecular Biophysics Protein Expression Facility for providing facilities and resources for protein expression; S. Miller of the FSU Sequencing facility for assistance with Sanger sequencing. We also thank the staff at LBMS, especially Guobin Hu, at PNCC, especially Nancy Meyer, and at Van Andel Institute Core for assistance with data collection. We acknowledge the use of the BSIR instrumentations, including the Titan Krios, funded by NIH grant S10 RR025080, the Vitrobot and Solaris Plasma Cleaner, supported by S10 RR024564, and the BioQuantum/K3, acquired through U24 GM116788 funding. We are also thankful to SECM4 (The Southeastern Center for Microscopy of Macro Molecular Machines) for screening and data collection through NIH grant R24 GM145964. This work was supported by NIH grant R35 GM152081 to H.L.

## Author contributions

**Charlisa Whyms**: Conceptualization; Investigation; Writing—original draft; Writing—review and editing. **Yu Zhao**: Data curation; Visualization; Methodology; Writing—original draft; Writing—review and editing. **Doreen Addo-Yobo**: Conceptualization; Data curation; Investigation; Writing—original draft; Writing—review and editing. **Huan He**: Data curation. **Arthur Carl Whittington**: Data curation. **Despoina Trasanidou**: Conceptualization. **Carl Raymund P Salazar**: Data curation; Investigation; Writing—review and editing. **Raymond H J Staals**: Conceptualization; Supervision; Writing—review and editing. **Hong Li**: Conceptualization; Supervision; Funding acquisition; Visualization; Writing—original draft; Project administration; Writing—review and editing.

Source data underlying figure panels in this paper may have individual authorship assigned. Where available, figure panel/source data authorship is listed in the following database record: biostudies:S-SCDT-10_1038-S44318-025-00578-y.

## Disclosure and competing interests statement

The authors declare no competing interests.

# Expanded View Figures

**Figure EV1.  Concentration effects by cA$_4$ and A$_2$ > P in their activation of *Taq*Cad1.**

"WT + ATP" indicates the reaction in the absence of any ligand. *Taq*Cad1-mediated reactions were dephosphorylated by calf intestinal alkaline phosphatase following heat deactivation. (**A**) UV absorption measurement results of the reaction of *Taq*Cad1 with ATP in the presence of various of cA$_4$ concentrations (0.2, 0.5, and 2 μM). (**B**) UV absorption measurement results of the reaction of *Taq*Cad1 with ATP in the presence of various of products of 20 μM cA$_4$ incubated with the K106A variant for 15, 30, and 60 min, respectively. (**C**) UV absorption measurement results of the reaction of *Taq*Cad1 with ATP in the presence of ring nuclease products resulted from incubating cA$_4$ at 5, 10, and 20 μM, respectively, with 2 μM WT *Taq*Cad1 for 30 (top) and 60 (bottom) min, respectively. For panels (**A–C**), data represent the mean ± standard deviation of the mean ($n = 3$ biological replicates). Source data are available online for this figure.

▶

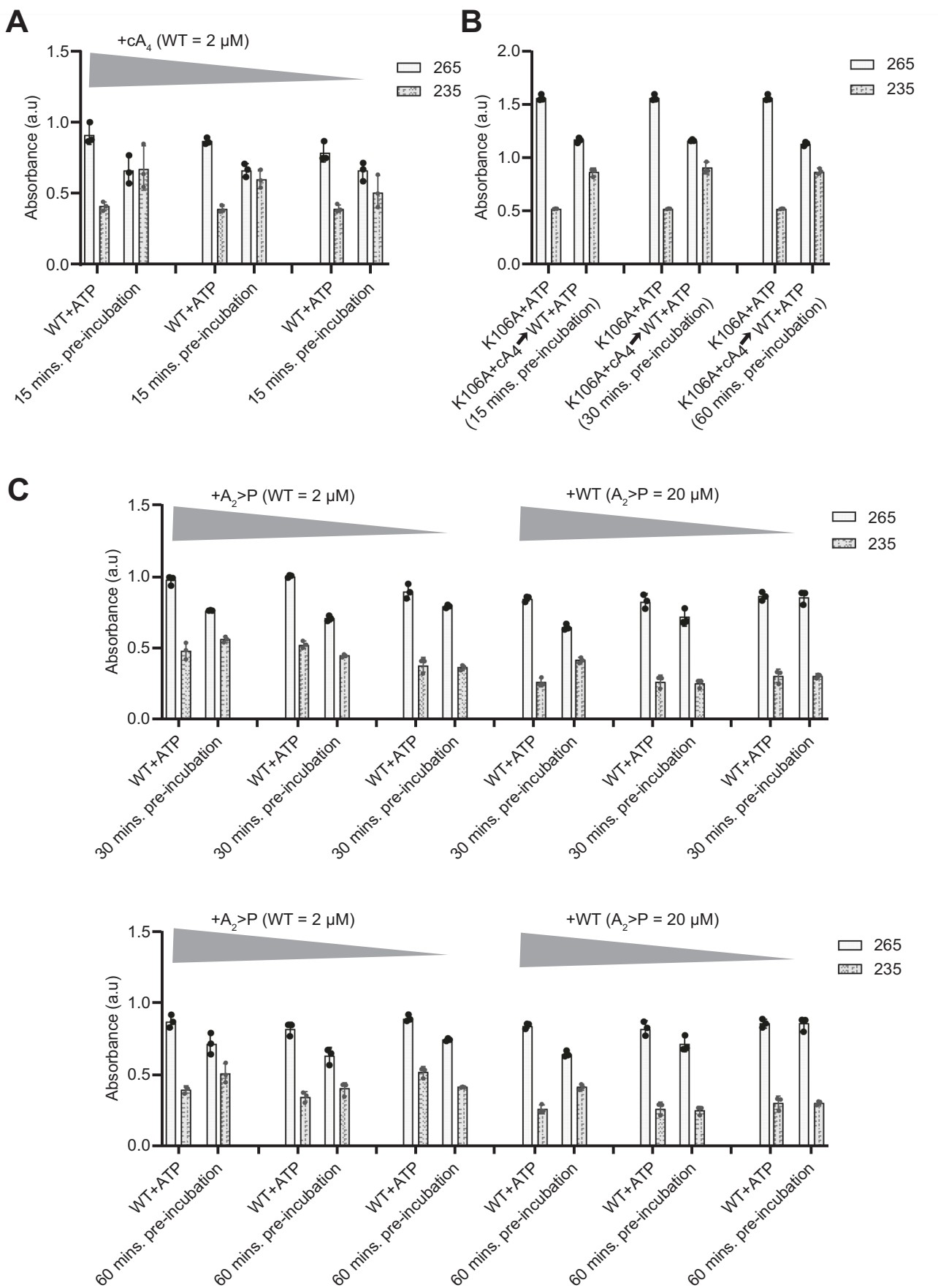

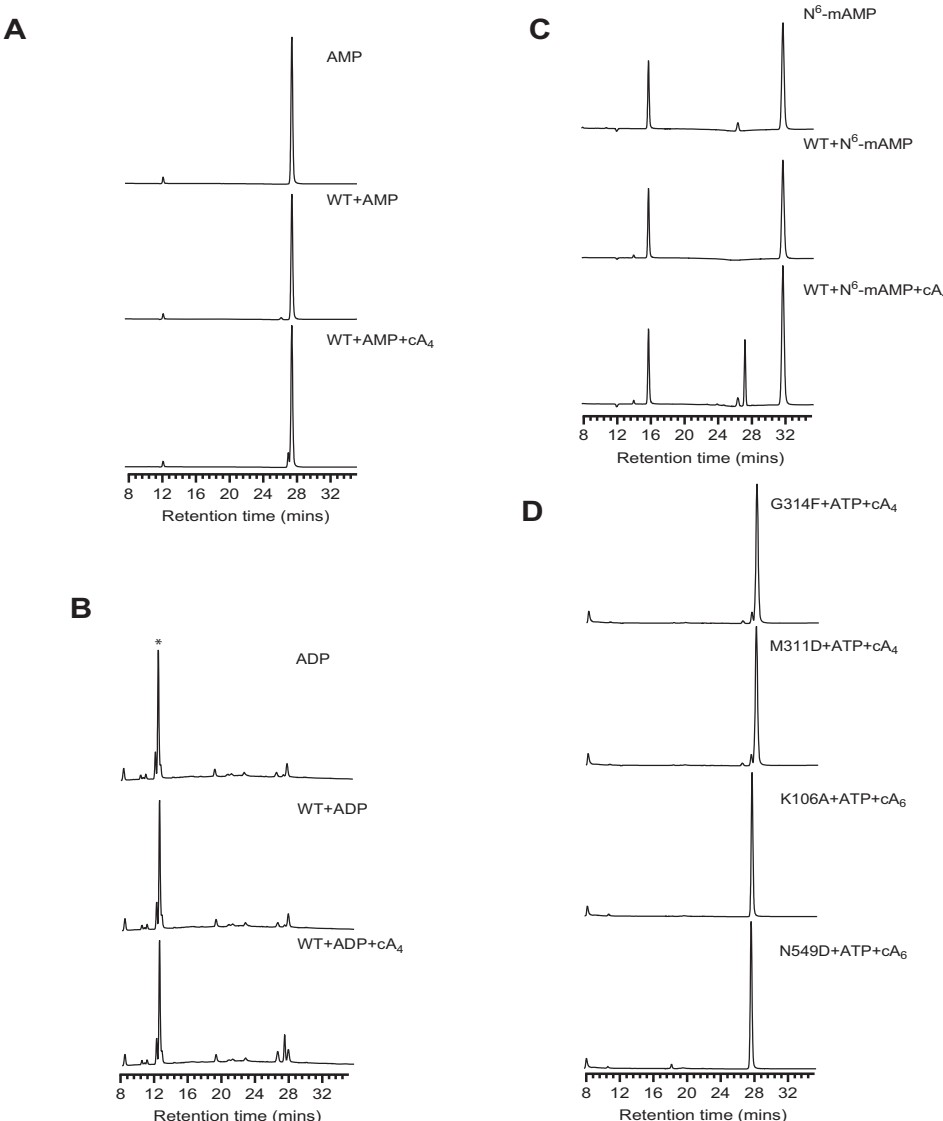

**Figure EV2.  Additional deamination assays for *Taq*Cad1.**

(A) HPLC profiles following *Taq*Cad1 reaction with adenosine monophosphate (AMP) in the presence and absence of cA$_4$. (B) HPLC profiles following *Taq*Cad1 reaction with adenosine diphosphate (ADP) in the presence and absence of cA$_4$. The asterisk indicates a nonspecific peak present in the unreacted ADP sample (top). (C) HPLC profiles following *Taq*Cad1 reaction with N$^6$-methyl-adenosine monophosphate (N$^6$m-AMP) in the presence and absence of cA$_4$. (D) HPLC profiles of the reaction products following incubating ATP with *Taq*Cad1 variants (G314F, M311D, N549D, or K106A) in the presence and absence of cA$_4$ or cA$_6$. The reaction products were treated with CIP. The profiles of adenosine and inosine are included for comparison. Dash lines indicate the elution positions of inosine and adenosine.

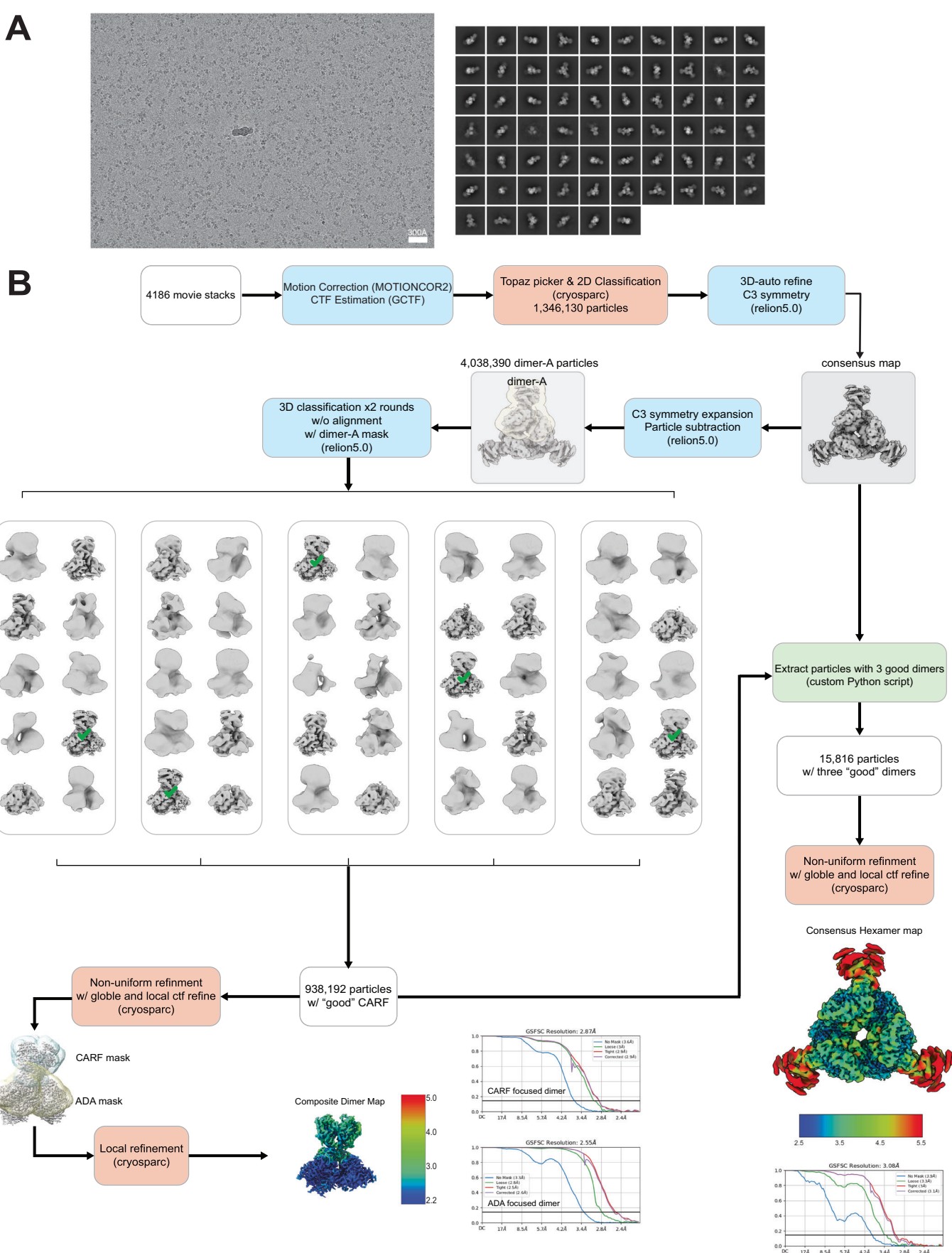

◀  **Figure EV3.  Data collection, processing, and 3D reconstruction of the cryoEM structures of apo *Taq*Cad1.**

(**A**) Example micrograph and 2D class averages (scale bar 300 Å). (**B**) Data collection, particle selection, and reconstruction flowchart. Two reconstructions were made for the individual dimers and the intact hexamer, respectively. The dimer was reconstructed from a C3 symmetry expansion, leading to over 4 million particles. 3D classification was performed to select the best dimers that were then reconstructed for the CARF and ADA domains, respectively, that were combined to form the composite map. The best hexamer containing three intact dimers was reconstructed without focused refinement. Final maps used for refinement are shown with local resolutions and their Fourier shell correlation (FSC) curves. 0.143 FSC cutoff was used for resolution estimation.

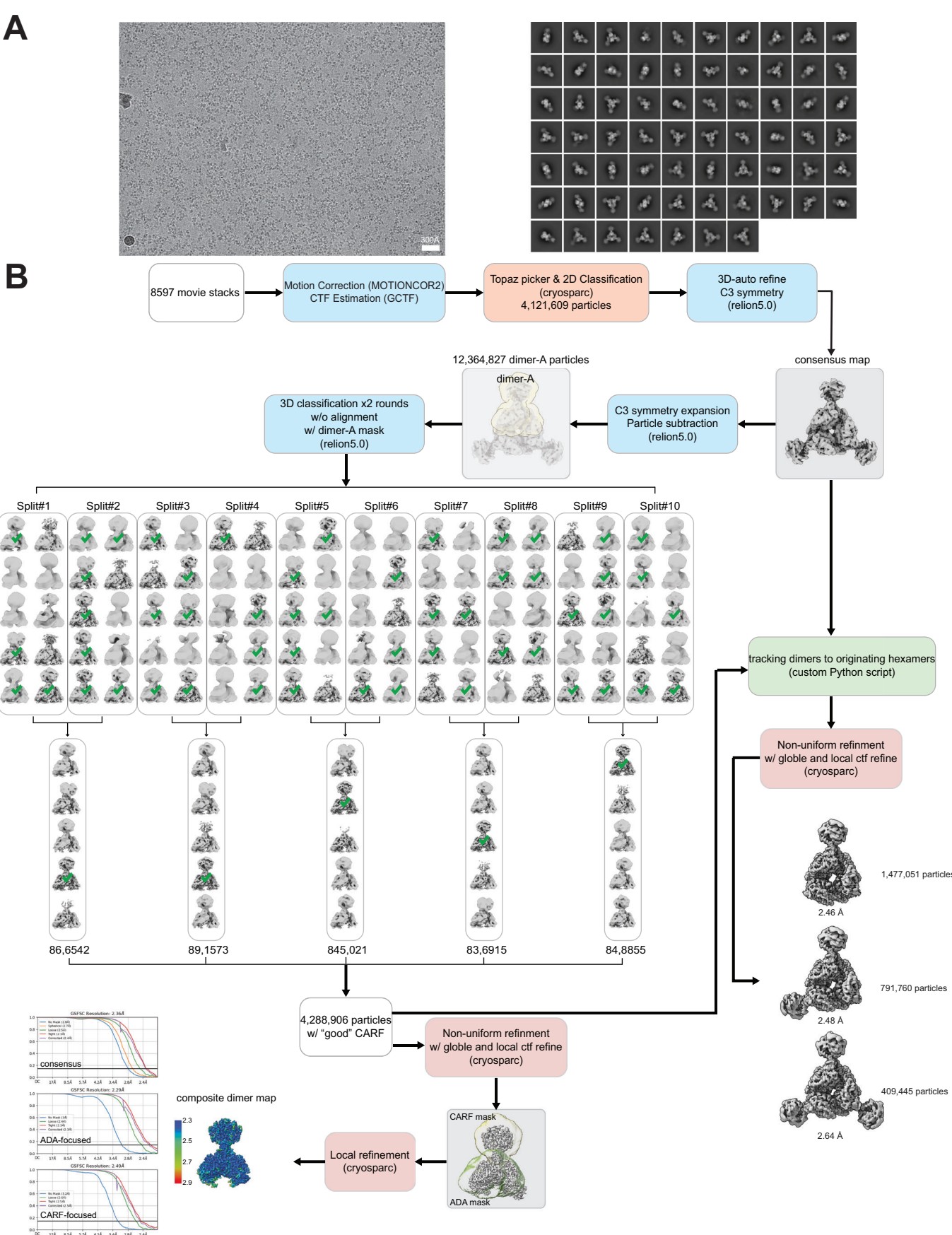

**Figure EV4. Data collection, processing, and 3D reconstruction of the cryoEM structures of *Taq*Cad1 incubated with cA$_4$.**

(A) Example micrograph and 2D class averages (scale bar 300 Å). (B) Data collection, particle selection, and reconstruction flowchart. Two reconstructions were made for the individual dimers and the intact hexamer, respectively. The dimer was reconstructed from a C3 symmetry expansion, leading to over 4 million particles. 3D classification was performed to select the best dimers that were then reconstructed for the CARF and ADA domains, respectively, that were combined to form the composite map. The hexamers containing one, two, or three intact dimers were reconstructed without focused refinement. Final maps used for refinement are shown with local resolutions and their Fourier shell correlation (FSC) curves. 0.143 FSC cutoff was used for resolution estimation.

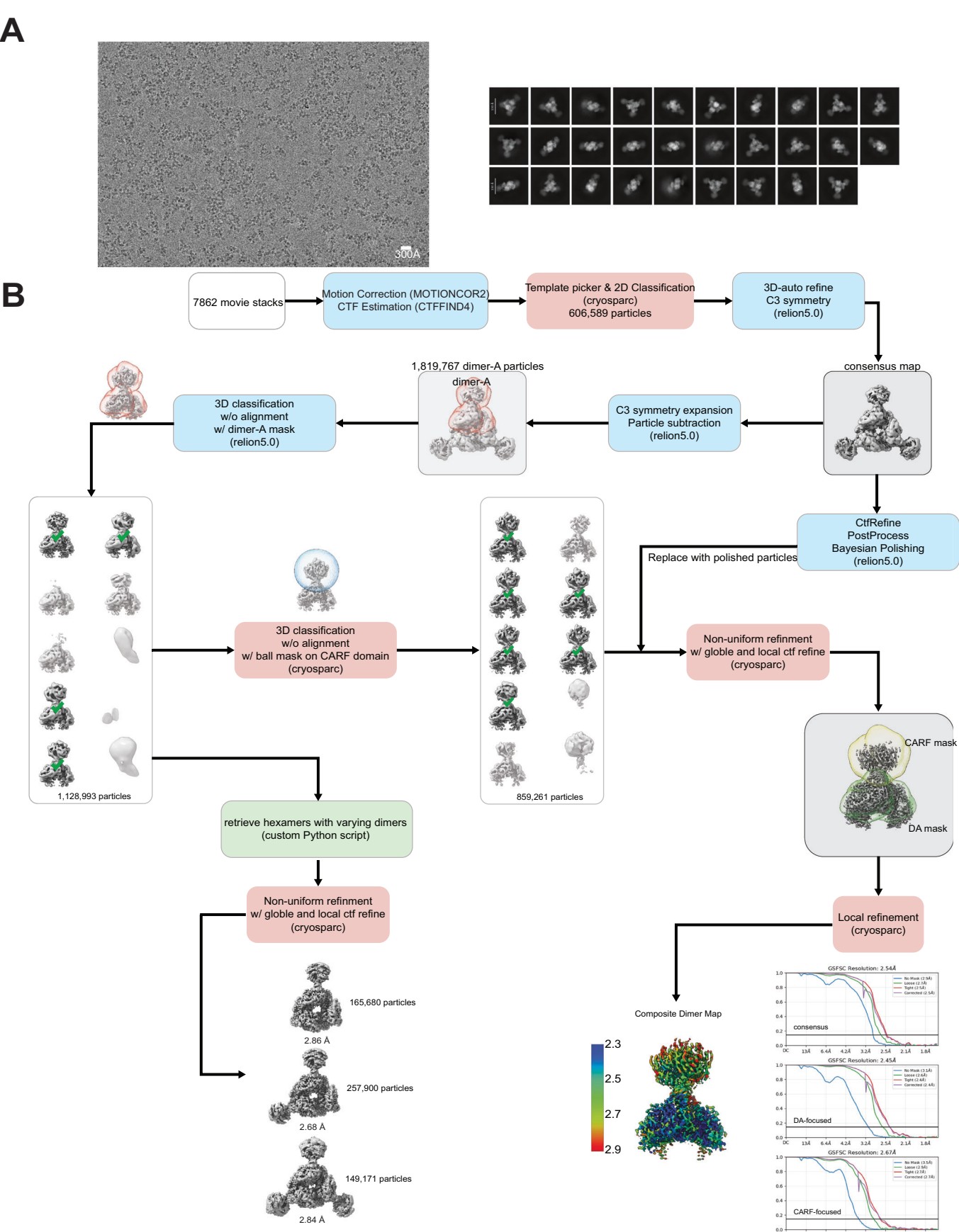

◄ **Figure EV5. Data collection, processing, and 3D reconstruction of the cryoEM structures of *Taq*Cad1 incubated with cA$_4$ and ATP.**

(A) Example micrograph and 2D class averages (scale bar 300 Å). (B) Data collection, particle selection, and reconstruction flowchart. Two reconstructions were made for the individual dimers and the intact hexamer, respectively. The dimer was reconstructed from a C3 symmetry expansion, leading to over 4 million particles. 3D classification was performed to select the best dimers that were then reconstructed for the CARF and ADA domains, respectively, that were combined to form the composite map. The hexamers containing one, two, or three intact dimers were reconstructed without focused refinement. Final maps used for refinement are shown with local resolutions and their Fourier shell correlation (FSC) curves. 0.143 FSC cutoff was used for resolution estimation.

