## [Peer Review File · The EMBO Journal]

The twist-and-squeeze activation of CARF-Fused Adenosine Deaminase by Cyclic Oligoadenylates

Charlisa Whyms, Yu Zhao, Doreen Addo-Yobo, Huan He, Arthur Whittington, Despoina Trasanidou, Carl Salazar, Raymond Staals, and Hong Li

Corresponding author(s): Hong Li (hong.li@fsu.edu)

Review Timeline:

Submission Date:	19th Dec 24
Editorial Decision:	10th Feb 25
Appeal Received:	22nd May 25
Editorial Decision:	6th Jun 25
Revision Received:	11th Jun 25
Editorial Decision:	14th Aug 25
Revision Received:	22nd Aug 25
Accepted:	13th Sep 25

Editor: Cornelius Schneider

Transaction Report:

Dear Prof. Li,

Thank you for submitting your manuscript for consideration by the EMBO Journal.

Your manuscript was sent to three referees and we have now received the comments of two of them. As the third reviewer has not yet returned his/her report and the other two are in fair agreement, I am taking a decision on your manuscript now, in order to save you from unnecessary delay. As you will see from the enclosed reports, neither referee supports publication of the manuscript in the EMBO Journal. I would prefer not to repeat their individual points of criticism in this letter, but both clearly point to a number of serious conceptual and technical shortcomings of your study, that, in our assessment, would preclude publication here.

Given these negative opinions and the fact that the EMBO Journal can only afford to accept papers which receive enthusiastic support from a majority of referees, we see little choice other to return the manuscript to you with the message that we cannot offer to publish it. I am sorry to have to disappoint you.

Yours sincerely,

Cornelius Schneider, PhD
Editor
The EMBO Journal
c.schneider@embojournal.org

Referee #1:

This manuscript presents a structural and biochemical study of TaqCad1, a CARF-domain containing protein fused with an adenosine deaminase (ADA) domain. The authors provides biochemical and structural analyses, suggesting that both cA4 and cA6 stimulate the ATP-to-ITP conversion of TaqCad1 through a hexameric assembly. Moreover, TaqCad1 appears to specifically degrade cA4 but not cA6. While the work is well-executed in many respects, the findings presented in the manuscript overlap considerably with previous studies (Li et al., Science, 2024 and Baca et al., Cell, 2024), particularly regarding the specificity for ATP, the cA4- and cA6-dependent activation, the cA4 cleavage activity within the CARF domain as well as the overall structural topology of the protein. Additionally, the manuscript's conclusions are based on incomplete structural data, which makes it difficult to fully assess the proposed activation mechanism. These issues reduce the novelty of the manuscript and suggest that additional work is needed to strengthen its overall impact.

Specific points

1. The authors show that in the cA4-bound structure, cA4 molecules are cleaved into A₂>P. This raises concerns that the observed structure may reflect an inactive form of TaqCad1, as previous studies have shown that cleavage of cA4 can lead to the inactivation of CARF-domain containing type III effectors (Li et al., Science, 2024; Jia et al, 2019; Du et al, 2024). To address this, I would recommend solving the cA6-bound structure, as cA6 is not degraded by TaqCad1. If this not feasible, the authors should carefully reconsider their interpretation of the cA4-bound structure to ensure it is consistent with the proposed activation mechanism.

2. The manuscript presents cryo-EM data for the cA4-bound structure, but the apo form of the CARF domain remains poorly resolved. The authors propose that cA4 binding allosterically activates the ADA domain, but without clear structural data for the apo form, it is difficult to assess the conformational changes that lead to activation. More complete structural data for the apo form would strengthen the argument for how cA4 binding induces the observed activation.

3. The previous publications have already discussed the ATP specificity, cA4-dependent activation, the ring nuclease activity, and hexameric structure of the homologous proteins in considerable detail. The manuscript does not sufficiently highlight how its findings extend beyond these earlier studies or contribute significantly new insights. I would recommend addressing this overlap more explicitly and clarifying the unique aspects of the study.

Line 150, Page 7: Please reference Bacteroidales Cad1 (Baca et al., Cell, 2024) and LngCAAD (Y. Li et al., Science, 2024), as these are more relevant to TaqCad1.

Referee #2:

Whymys et al report a biochemical and structural study of a CRISPR Cad1 protein from *Thermus aquaticus*, involved in type III crisper antiviral immunity. Cad1 is shown to bind and degrade cA4, and to possess cA4-dependent ATP deaminase activity. Weaker activation is seen with cA6. The structure of Cad1 reveals a trimer of dimers with cA4 (or its degradation products) bound in a dimeric CARF domain. Modest anti-plasmid activity can be detected in vivo. In general, there is a lack of experimental

detail and in particular a lack of clear explanation of the statistical treatment of data.

The authors were unfortunately scooped by a prior publication (Baca et al., Cell, 2024). This paper cannot be ignored and must be taken into account throughout the manuscript.

Main points:

1. The work needs to be put in the proper context by citing and fully discussing the work of Baca et al (Cell, 2004) on the same enzyme. Text must be revised throughout the manuscript to take account of this publication (see some points below).
2. Figure 1C. How were the peaks corresponding to A₂>P and A>P defined? Suppl figure 1D leaves many details unclear, making it impossible for the reader to interpret this experiment.
3. Figure 1D. This is used as evidence that Cad1 regulates the activity of Csx1. The reactions contained "varying concentrations of Cad1" but these are not defined. Only one gel is presented and replicates are not mentioned. Overall, this is weak evidence that Cad1 regulates the activity of Csx1 - other than in an in vitro experiment.
4. For figure 2, there is no mention of the number of replicates or statistical treatment. It looks like only duplicate measurements were made, which is insufficient for statistical analysis - error bars shown here (which are not explained) are inappropriate.
5. The structure of TaqCad1 should be properly compared to the structure of Bacteroidales Cad1 (Baca et al.) to place the work in a proper context. For example, was ATP identified bound at the inter-domain interfaces in this structure? How do the structural changes that accompany cA4 binding compare in the two structures?
6. In figure 3, it is surprising that Inosine co-elutes with ITP and adenosine with ATP, given the very different charges of these molecules. This is exemplified by the very different elution time of ADP (Figure S2). Do adenosine and ATP really co-elute at 27 min when ADP elutes at 13 min? How do the authors explain this? Furthermore, no mention is made here of experimental replicates.

Specific points:

1. Abstract: "Cad1 has a yet to be determined function". In fact this function was determined and published by Baca et al. Cell 2004. This makes the statement "revealing its unexpected role in deaminating ATP..." also incorrect/out of date. Please revise the text of the abstract accordingly.
2. Line 50 please revise text to clarify that RNA binding (and not degradation) triggers the Cas10 enzyme.
3. Line 56. CBASS does not produce diverse cOAs but rather diverse cyclic nucleotides.
4. Line 87. Hard to say this observation is surprising, given the prior art on this enzyme.
5. Line 91 "defending viruses" should be "defending against viruses".
6. It is interesting that TaqCad1 is specific for cA4 whereas Cad1 from Bacteroidales binds both cA4 and cA6 (Baca et al 2024 - not cited). This should be discussed and the work put in proper context.
7. Line 112 typo "mass spectroscopy" should be spectrometry.
8. Line 115 typo "K106A" should be "the K106A variant"
9. Line 257 typo "did not have cause"

** As a service to authors, EMBO Press provides authors with the possibility to transfer a manuscript that one journal cannot offer to publish to another EMBO publication or the open access journal Life Science Alliance launched in partnership between EMBO Press, Rockefeller University Press and Cold Spring Harbor Laboratory Press. The full manuscript and if applicable, reviewers' reports, are automatically sent to the receiving journal to allow for fast handling and a prompt decision on your manuscript. For more details of this service, and to transfer your manuscript please click on Link Not Available. **

Dear Dr. Schneider,

We are confident that we have compelling grounds for resubmitting our work previously reviewed at *EMBO J*, now with a revised title “**Sequential Conformational Activation of a CARF-Fused Adenosine Deaminase by Cyclic Oligoadenylates**”.

The major criticism from both reviewers in the previous review include:

- (1) incomplete structural analysis
- (2) not incorporating two recently published work about the homologous Cad1 enzymes
- (3) the use of native ligand cA₄ resulting in its cleavage in the complexes.

Briefly, we have obtained and analyzed a new cryoEM data set of the wild-type TaqCad1 incubated with cA₄ (the native ligand) and ATP, making available structures of all three functional states: apo, bound with cA₄ (or its cleavage product, A₂>P), and bound with cA₄ (or its cleavage product, A₂>P) and ATP (or its deaminated product) (concern #1). The new data and analysis experimentally validated the classic sequential activation model as identified by Koshland, D.E., Némethy, G., & Filmer, D. (*Biochemistry*, 1966, 5(1), 365–385.), an important novel insight on the activation mechanism not revealed by either of the two prior publications. We completely revised the manuscript in the context of the new data as well as the two recent publications mentioned by the reviewers (concern #2). Finally, we extensively characterized and confirmed the ability for the cleaved product of cA₄, primarily A₂>P, to trigger TaqCad1 activation (Figure 1c, Figure 2c, and Supplementary Figure 2) (concern #3). *We believe that the revised manuscript now offers a timely and comprehensive analysis of this class of enzyme, with particular emphasis on its molecular mechanism of activation.* Our understanding of, and responses to, each major criticism are detailed below.

Concern #1. “incomplete structural analysis”. We thank reviewers’ suggestions and agree our initial analysis was limited. We were only able to present structural analysis of the apo and the cA₄ (or its cleavage product A₂>P)-bound complexes (TaqCad1-cA₄ hereafter) and lack that for important TaqCad1-cA₄-ATP complex during the initial submission. We also did not completely characterize the apo structures due to its extreme heterogeneity. Notably, an incomplete apo structure was also achieved for LngCAAD by Li et al. *Sci.* 2025 (PDB code: 8Z40), likely due to the similar heterogeneity. With the new TaqCad1-cA₄-ATP data set, we devised a unique cryoEM single particle analysis method (figure below), analogous to that by Roh et al. *PNAS*, 2017 in studying GroEL subunit conformational variations, that leverages the symmetry of the TaqCad1 and re-analyzed all three complexes including the challenging apo complex. For each functional complex, we were able to reconstruct a high-quality dimer structure and various hexamer structures. The best dimers of the three functional states allowed us to quantify conformational changes with respect to the ADA domain including its active site, leading to the distinct structural features for defining the Relaxed (R) and the Taut (T) conformations, respectively (figure below).

Our careful analysis of the three high-quality dimer structures also identified a dynamic protein network, linking the cA_4 binding site to the ADA domain, that underlies the R-to-T conformational transition within the dimer (Figure 6), similar to that observed in LngCAAD (Li et al., 2025). We also consistently observed a sidechain rotation of the catalytic His461 upon cA_4 binding, similar to that observed in BabCad1 (Baca *et al.*, 20204). However, the two important structural changes were not simultaneously reported in either publication. Our work, therefore, provides a more complete characterization (Figure 6 & figure above).

We also performed careful analysis of the hexamer structures. We were able to resolve more complete apo hexamer structure than that for LngCAAD (Li et al. 2025). Importantly, we showed that in the hexamers containing one or two intact dimers, the ADA domains linked to CARF with a bound cA_4 are in the T states whereas those linked to cA_4 -free CARF adopt a hybrid R and T states (R/T hybrid), some of which are able to bind ATP (Figure 7 & figure below). Though Li et al. 2025 also saw heterogenous classes with different dimers by canonical 3D classifications, they did not model them, thus providing no experimental evidence for the sequential activation model. We thus believe that our analysis provides strong structural evidence for cooperativity and distinguishes its sequential activation model as identified by Koshland, D.E., Némethy, G., & Filmer, D. (*Biochemistry*, 1966, 5(1), 365–385.) from the concerted model by Monod, J., Wyman, J., & Changeux, J.-P. (*Journal of Molecular Biology*, 1965, 12, 88–118.). This conclusion is uniquely insightful to the broad CARF family of proteins.

To reflect our unique analysis, we revised the title of the paper to “**Sequential Conformational Activation of a CARF-Fused Adenosine Deaminase by Cyclic Oligoadenylates**”.

Concern #2. “not incorporating two recently published work”. We thank reviewers’ suggestions. We were unable to incorporate the findings from the two published studies in our initial submission due to time constraints. In this revised manuscript, we have thoroughly updated our analysis to include and address the results of these previous publications. We believe that our structural comparisons not only help resolve the controversy raised by these studies but also provide a broadly applicable model of activation for Cad1 enzymes.

We state the disparate mechanisms in the revised Introduction by the two studies: “Interestingly, the two studies differ in the mechanism of activation. Upon cOA binding, whereas a significant conformational change was observed in LngCAAD (Li *et al.*, 2025), a subtle sidechain adjustment of a single ADA catalytic histidine of BabCad1 was detected (Baca *et al.*, 2024). Based on the conformational changes and the observed Hill coefficient of deamination, the study of LngCAAD presents a model of cooperative regulation, however, with limited structural support (Li *et al.*, 2025). In addition, the two studies disagree on the roles of Mg²⁺, whereas Mg²⁺ is believed to facilitate ATP binding to LngCAAD (Li *et al.*, 2025), it is believed to participate in catalysis in BabCad1 (Baca *et al.*, 2024). Thus, the molecular mechanism underlying cOA-mediated activation remains unclear.”

Our analysis helps to address the two seemingly different modes of structural transitions (large conformational vs subtle sidechain rotation) by clearly identifying both in TaqCad1 (Figures 5-7). In the Discussion section, we propose possible reasons for the different modes and raise the question if this difference is due to mechanistic diversity or limitation in current methods.

We state in the Discussion that “The large conformational changes in *TaqCad1* upon cA₄ binding resemble those observed in *LngCAAD* (Li *et al.*, 2025). Interesting, the more subtle but convincing sidechain rotation of the catalytic His461 resembles was observed in *BabCad1* that, interestingly, did not exhibit gross conformational changes (Baca *et al.*, 2024). We propose that this observation underscores the structural and mechanistic diversity of CARF enzymes, which employ distinct conformational dynamics to enable activation. Notably, certain conformational changes—such as those in *BabCad1*—appear undetectable with the current methods. This raises intriguing questions: Are these CARF proteins capable of cooperative regulation? If not, could this limitation arise from their unique physiological context rather than intrinsic molecular constraints?”

We also were able to model the Mg²⁺ ions unambiguously and revealed their roles in ATP binding, as identified by Li *et al.*, 2025, but not in catalysis as reported by Baca *et al.* 2024. We believe that this is an important mechanistic addition to the study of Cad1 enzymes.

In summary, although our study partially overlaps with the two previously published works, it offers a more advanced analysis of the structural data, culminating in a cooperative activation model supported by sequential conformational changes. Structural comparisons between our structures with those previously published are presented in Supplementary Figure 10 and the figure on the next page.

a

BabCad1 (Baca et al. 2024)

LngCAAD (Li et al. 2025)

**b**

TaqCad1 (This work)

**c**

BabCad1 vs TaqCad1

LngCad1 vs TaqCad1

Structural comparison across conformational states and among the three studies indicate that binding of the weaker activator, cA₆ for LngCAAD and A2>P, leads to the same activated Cad1 conformations.

Concern #3. “the use of native ligand cA₄”. We thank reviewers’ suggestions. We agree that our initial submission did not address if the ring nuclease product A₂>P can activate TaqCad1, and if so, at what conditions.

We first would like to point out that in the study by Li et al. Sci. 2025 on LngCAAD, authors showed that the ring nuclease products, also primarily A₂>P, robustly trigger ATP-to-ITP conversion (Figure 1E and Supplementary Figure 10), supporting that A₂>P-bound LngCAAD is in an activated state.

We addressed the same question independently. In the revised manuscript, we performed extensive analysis of the deamination activity triggered by the ring nuclease products, primarily A₂>P (Figure 2c and Supplementary Figure 2, also see figure below). We pre-incubated cA₄ with the wild-type TaqCad1 at various time intervals to ensure cleavage of the cA₄ and then applied the cleaved products at various concentrations to activate fresh deamination reactions by TaqCad1. The figure below shows one of the three cA₄ cleavage time intervals (15 minutes) we tested. Indeed, while low concentrations of A₂>P did not, high concentrations of A₂>P readily activated deamination. Other two cleavage time intervals (30 mins and 60 mins) showed the same results. As expected, the same concentration threshold effect was also observed for the native ligand cA₄ except for at a much lower concentration range than A₂>P, consistent with its nanomolar affinity for TaqCad1 (figure below). Consistently, reducing TaqCad1 enzyme concentration while maintaining that of A₂>P also reduced deamination. Furthermore, the fact that ATP are readily bound to the A₂>P-trapped TaqCad1 also supports its ability to activate the enzyme (Figures 4-7). Therefore, similar to the other weaker ligand, cA₆, A₂>P can bind and activate TaqCad1, supporting that our A₂>P bound structures are indeed activated structures.

Our extensive structure comparison with those by Li et al. 2025 further supports that A₂>P-bound TaqCad1 is activated. There is high structural homology between our A₂>P- or A₂>P- and ATP-bound structures and that of LngCAAD bound with cA₆, its weaker ligand (Supplementary Figure 10 & the figure on the previous page). Both weaker ligands can activate Cad1 deamination activities if they are able to bound with the enzyme at elevated concentrations.

We have indicated this important conclusion in the Results section.

Responses to specific concerns raised by both reviewers are highlighted.

Referee #1:

This manuscript presents a structural and biochemical study of TaqCad1, a CARF-domain containing protein fused with an adenosine deaminase (ADA) domain. The authors provides biochemical and structural analyses, suggesting that both cA4 and cA6 stimulate the ATP-to-ITP conversion of TaqCad1 through a hexameric assembly. Moreover, TaqCad1 appears to specifically degrade cA4 but not cA6. While the work is well-executed in many respects, the findings presented in the manuscript overlap considerably with previous studies (Li et al., Science, 2024 and Baca et al., Cell, 2024), particularly regarding the specificity for ATP, the cA4- and cA6-dependent activation, the cA4 cleavage activity within the CARF domain as well as the overall structural topology of the protein. Additionally, the manuscript's conclusions are based on incomplete structural data, which makes it difficult to fully assess the proposed activation mechanism. These issues reduce the novelty of the manuscript and suggest that additional work is needed to strengthen its overall impact.

Responses: We thank the reviewer for providing the constructive criticism and acknowledging our experimental approach and findings. We understand the need for additional experimental data and incorporation of the two recent findings on the homologous Cad1 enzymes (Li et al. Sci. 2025 and Baca et al, Cell, 2024). We agree that a timely comparison of ours and the two recently published works is highly valuable to the field.

In the revise manuscript, we fully incorporated these two works throughout all sections. We leveraged our now complete structural data sets of the three functional states (Figures 4-7) in identifying the similarities and differences both within and across the studies. A complete structural alignment is provided in Supplementary Figure 10 (also see figure under concern #3). This analysis, while acknowledging the value of each study, advances the mechanistic understanding of the allosteric activation mechanism.

Specific points

1.The authors show that in the cA4-bound structure, cA4 molecules are cleaved into $A_2>P$. This raises concerns that the observed structure may reflect an inactive form of TaqCad1, as previous studies have shown that cleavage of cA4 can lead to the inactivation of CARF-domain containing type III effectors (Li et al., Science, 2024; Jia et al, 2019; Du et al, 2024). To address this, I would recommend solving the cA6-bound structure, as cA6 is not degraded by TaqCad1. If this not feasible, the authors should carefully reconsider their interpretation of the cA4-bound structure to ensure it is consistent with the proposed activation mechanism.

Responses: We thank the reviewer for this comment and the citations provided. We agree that we should have characterized the role of the ring nuclease products, primarily $A_2>P$, in enzyme activation to support our structural observations. To this end, we performed extensive analysis

of the deamination activity triggered by the ring nuclease products (Figure 2c and Supplementary Figure 2, also see the figure under concern #3). We can convincingly conclude that $A_2>P$ is able to trigger deamination if present at an elevated concentration to enable its binding. This conclusion resulted from functional assays agrees with our new cryoEM structures where high cA_4 concentrations were used, demonstrating that the $A_2>P$ trapped TaqCad1 is fully bound with ATP (Figures 4-7, see also the 2nd figure under concern #1).

The finding that elevated $A_2>P$ levels activate CARF enzymes does not conflict with the findings cited by the reviewers. In one of these studies (Jia et al. Mol. Cell 2019) where the concentration effect was tested, Csm6 (cA_4 -activated nuclease) could be activated when its concentration was elevated using synthetic $A_2>P$ (Figure 5A).

We believe our functional characterization with $A_2>P$ and its activated TaqCad1 structures complement the study by Li et al. 2025 where the weaker cA_6 was used in structural analysis. We do not know, though, if the structural differences between our $A_2>P$ - and their cA_6 -bound structures, especially in sequential conformational transitions (Figure 7), are a result of the different ligands or the cryoEM analysis methods. This specific question may be addressed in additional studies.

2.The manuscript presents cryo-EM data for the cA_4 -bound structure, but the apo form of the CARF domain remains poorly resolved. The authors propose that cA_4 binding allosterically activates the ADA domain, but without clear structural data for the apo form, it is difficult to assess the conformational changes that lead to activation. More complete structural data for the apo form would strengthen the argument for how cA_4 binding induces the observed activation.

Responses: We agree that the initial structural analysis of the apo form can be improved despite the challenges in heterogeneity. We adopted a single particle analysis procedure previously used for analyzing GroEL subunit conformational heterogeneity (Roh et al. PNAS, 2017, 114(31) 8259-8264) to achieve the best possible resolution of the individual dimers while revealing the extent of the heterogeneity for all complexes including the apo complex. This procedure allowed us to obtain a high-quality dimer structure and a good hexamer structure for each functional state. As a result, we were able to characterize structural changes up cA_4 binding (Figures 5-7, also see the two figures under concern #1).

3.The previous publications have already discussed the ATP specificity, cA_4 -dependent activation, the ring nuclease activity, and hexameric structure of the homologous proteins in considerable detail. The manuscript does not sufficiently highlight how its findings extend beyond these earlier studies or contribute significantly new insights. I would recommend addressing this overlap more explicitly and clarifying the unique aspects of the study.
Line 150, Page 7: Please reference Bacteroidales Cad1 (Baca et al., Cell, 2024) and LngCAAD (Y. Li et al., Science, 2024), as these are more relevant to TaqCad1.

Responses: We thank the reviewer for this comment. With the now available ATP-bound structures and our unique analysis, we uncovered a sequential conformational change

previously unavailable (Figure 7). This significant finding is a key to the previously proposed cooperative activation model by Li et al. 2025.

Other important and unique contributions include identification of the dynamic protein network supported by hexameric assembly and its specific role in remodeling helix $\alpha 21'$ and the catalytic residue, His461 (Figures 4-6).

Referee #2:

Whymys et al report a biochemical and structural study of a CRISPR Cad1 protein from *Thermus aquaticus*, involved in type III crisper antiviral immunity. Cad1 is shown to bind and degrade cA4, and to possess cA4-dependent ATP deaminase activity. Weaker activation is seen with cA6. The structure of Cad1 reveals a trimer of dimers with cA4 (or its degradation products) bound in a dimeric CARF domain. Modest anti-plasmid activity can be detected in vivo. In general, there is a lack of experimental detail and in particular a lack of clear explanation of the statistical treatment of data.

Responses: We thank the reviewer for providing thoughtful criticism and a positive overview of our initial submission. The revised manuscript has significantly expanded in experimental details and data analysis.

The authors were unfortunately scooped by a prior publication (Baca et al., Cell, 2024). This paper cannot be ignored and must be taken into account throughout the manuscript.

Responses: We appreciate the understanding of the reviewer. In addition to Baca et al. Cell, 2024, another publication by Li et al. Sci. 2025 also described findings of a homologous Cad1. However, while sharing some key findings, the two studies presented several mechanistic discrepancies. In addition, the Li et al. Sci. 2025 presented an intriguing model of activation with limited structural support.

In our revised manuscript, we present a set of cryoEM structures of TaqCad1 at the apo and activated state, thereby identify structural features supporting a sequential conformational activation model (Figures 4-7, and see figures under concern #1).

Main points:

1. The work needs to be put in the proper context by citing and fully discussing the work of Baca et al (Cell, 2004) on the same enzyme. Text must be revised throughout the manuscript to take account of this publication (see some points below).

Responses: We thank the reviewer for pointing out the publication by Baca et al. Cell, 2024 (and Li et al. Sci. 2025). We agree that incorporation of these two recent findings on the homologous Cad1 enzymes would strengthen our work.

In the revised manuscript, we fully incorporated these two works throughout all sections. We leveraged our now complete structural data sets of the three functional states (Figures 4-7) in identifying the similarities and differences both within and across the studies. A complete structural alignment is provided in Supplementary Figure 10. This analysis, while acknowledging the value of each study, advances the mechanistic understanding of the allosteric activation mechanism.

2. Figure 1C. How were the peaks corresponding to A2>P and A>P defined? Supplementary figure 1D leaves many details unclear, making it impossible for the reader to interpret this experiment.

Responses: We thank the reviewer for pointing out the neglected raw data. We identified all possible cleavage products based on mass spectrometry analysis derived from the matching m/z values (see figure below). We will include the raw mass spectrometry peaks corresponding to these products in the revised Supplementary Figure 1d.

3. Figure 1D. This is used as evidence that Cad1 regulates the activity of Csx1. The reactions contained "varying concentrations of Cad1" but these are not defined. Only one gel is presented and replicates are not mentioned. Overall, this is weak evidence that Cad1 regulates the activity of Csx1 - other than in an in vitro experiment.

Responses: We thank the reviewer for this comment. We now clearly state the concentrations used in Figure 1D and added another set of reactions when cA₄ concentration was increased. We now include a Supplementary Data File that includes multiple replicates of this experiment.

The reviewer raised an interesting question about if Csx1 and Cad1 function together in vivo. To fully address this question, a rigorous in vivo assay system, preferably in the native organism, is needed, which we believe is beyond the scope of this study.

4. For figure 2, there is no mention of the number of replicates or statistical treatment.

It looks like only duplicate measurements were made, which is insufficient for statistical analysis - error bars shown here (which are not explained) are inappropriate.

Responses: We thank the reviewer for this comment. We now include multiple replicates of the assays in the new Supplementary Data File. We also explicitly stated the error calculations in Figure 2 caption.

5. The structure of TaqCad1 should be properly compared to the structure of Bacteroidales Cad1 (Baca et al.) to place the work in a proper context. For example, was ATP identified bound at the inter-domain interfaces in this structure? How do the structural changes that accompany cA₄ binding compare in the two structures?

Responses: We thank the reviewer for this comment. We have now compared the TaqCad1 structures to those by Baca et al. and Li et al. (Supplementary Figure 10 and also see the figure under concern #2). Like those by Li et al., we found no evidence for the inter-domain ATP. We also extensively characterized, given our unique cryoEM analysis method and the new ATP-bound data set, the conformational changes accompanying cA₄ binding and how they differ from those reported by two previous publications.

6. In figure 3, it is surprising that Inosine co-elutes with ITP and adenosine with ATP, given the very different charges of these molecules. This is exemplified by the very different elution time of ADP (Figure S2). Do adenosine and ATP really co-elute at 27 min when ADP elutes at 13 min? How do the authors explain this? Furthermore, no mention is made here of experimental replicates.

Responses: We thank the reviewer for this comment. We should emphasize that this assay makes use of calf intestinal alkaline phosphatase that dephosphorylates nucleotides to adenosine or inosine, explaining the overlapping elution profiles. We now more clearly state this fact in relevant sections and figure captions. For the ADP profile, we apologize for including the explanation that "The asterisk indicates non-specific peak present in the unreacted ADP sample (top)". The adenosine still elutes around 27 min. We also include experimental replicates in the Supplementary Data File.

Specific points:

1. Abstract: "Cad1 has a yet to be determined function". In fact this function was determined and published by Baca et al. Cell 2004. This makes the statement "revealing its unexpected role in deaminating ATP..." also incorrect/out of date. Please revise the text of the abstract accordingly.

Responses: We thank the reviewer for this comment. We now revised the abstract to include the phrase "(Cad1) and has recently been shown to convert ATP to ITP."

2. Line 50 please revise text to clarify that RNA binding (and not degradation) triggers the Cas10 enzyme.

Responses: We thank the reviewer for this comment. We revised this statement.

3. Line 56. CBASS does not produce diverse cOAs but rather diverse cyclic nucleotides.

Responses: We thank the reviewer for this comment. We revised the phrase to cyclic oligonucleotides.

4. Line 87. Hard to say this observation is surprising, given the prior art on this enzyme.

Responses: We thank the reviewer for this comment. We removed the phrase “surprising”.

5. Line 91 "defending viruses" should be "defending against viruses".

Responses: We thank the reviewer for this comment. This statement is no longer in the revised text.

6. It is interesting that TaqCad1 is specific for cA4 whereas Cad1 from Bacteroidales binds both cA4 and cA6 (Baca et al 2024 - not cited). This should be discussed and the work put in proper context.

Responses: We thank the reviewer for this comment. Given the new publication by Li et al. Sci. 2025, we know Cad1 also has diverse ligands. In our study, we thoroughly addressed the ability of cA₄, cA₆, and A₂>P to trigger the activity by including binding and enzymatic assays (Figure S1b, Figure 2c and Figure S2).

7. Line 112 typo "mass spectroscopy" should be spectrometry.

Responses: We thank the reviewer for this comment. We revised the phrase.

8. Line 115 typo "K106A" should be "the K106A variant"

Responses: We thank the reviewer for this comment. We revised the phrase.

9. Line 257 typo "did not have cause"

Responses: We thank the reviewer for this comment. We removed “have”.

Dear Prof. Li,

Thank you for submitting your manuscript for consideration by the EMBO Journal. It has now been seen by three referees whose comments are enclosed. As you will see, both referees express interest in your manuscript and are broadly in favour of publication, pending satisfactory minor revision.

Given the referees' positive recommendations, I would like to invite you to submit a revised version of the manuscript, addressing the remaining comments of referee #2.

Thank you for the opportunity to consider your work for publication. I look forward to your revision.

Yours sincerely,

Cornelius Schneider, PhD
Editor
The EMBO Journal
c.schneider@embojournal.org

Use the link below to submit your revision:

Referee #1:

The authors have addressed all my concerns. I recommend publication of the revised version.

Referee #2:

Whyms et al. provide a significantly revised version of their manuscript on the Cad1 effector, including substantial new data and placing the work in its proper context. The concerns of both reviewers have been dealt with thoughtfully, resulting in a much improved manuscript.

The main remaining weakness is the in vivo data shown in figure 8, which shows little if any effect of Cad1 activation. This may be due to the sub-optimal temperature used, but the in vitro activity measurements were carried out at the same lower temperatures. If space is an issue, figure 8 could be deleted or moved to supplemental.

Specific points:

1. Line 114. It is unclear what constitutes "a novel role" here. Words such as "novel" are best avoided.
2. Line 128. Where the error is about 50% it is not advisable to quote 3 significant figures for the Kd.
3. Line 142. The mechanism of metal independent ring nuclease activity has already been described for other CARF family proteins, which is perhaps more relevant than a comparison to RNaseA. These studies should be cited here.
4. Line 282 What does "well served" mean in this context?
5. Line 347. Discussion
6. Line 376. The suggestion that "oligomerization uniquely evolved in TaqCad1" is misleading - it is probably conserved in the broader family.
7. Line 376. The term "protein network" is imprecise, suggesting multiple proteins.
8. Line 382. Interestingly
9. Line 383. "resemble was observed" is not grammatically correct.
10. Line 389. What is meant by "unique physiological context" here?
11. Figure 8. What is "standard deviation of the mean"? Is this SD or SEM? For triplicate data points, SD should be used.
12. Figure 8C is not really strengthening the paper, since it seems the empty vector control has a clearer phenotype than TaqCad1. Probably worth removing this.

Referee #1:

The authors have addressed all my concerns. I recommend publication of the revised version.

Referee #2:

Whyms et al. provide a significantly revised version of their manuscript on the Cad1 effector, including substantial new data and placing the work in its proper context. The concerns of both reviewers have been dealt with thoughtfully, resulting in a much improved manuscript.

The main remaining weakness is the in vivo data shown in figure 8, which shows little if any effect of Cad1 activation. This may be due to the sub-optimal temperature used, but the in vitro activity measurements were carried out at the same lower temperatures. If space is an issue, figure 8 could be deleted or moved to supplemental.

Responses: We agree with the reviewer that the mild in vivo effects of TaqCad1 could be due to reasons not related to our in vitro findings. We also agree that moving Figure 8 to supplementary materials is appropriate.

Specific points:

1. Line 114. It is unclear what constitutes "a novel role" here. Words such as "novel" are best avoided.

Responses: We removed the entire statement.

2. Line 128. Where the error is about 50% it is not advisable to quote 3 significant figures for the Kd.

Responses: Thank the reviewer for catching this error. We now report Kd with a single significant figure consistent with the standard errors.

3. Line 142. The mechanism of metal independent ring nuclease activity has already been described for other CARF family proteins, which is perhaps more relevant than a comparison to RNaseA. These studies should be cited here.

Responses: We thank the reviewer for pointing this negligence out. We added the references of Niewoehner et al. 2016; Athukoralage et al., 2018; and Jia et al. 2019 that all demonstrated the metal independence to the phrase “As shown for other ring nucleases”.

4. Line 282 What does "well served" mean in this context?

Responses: We thank the reviewer for pointing this out. The phrase should have been “well-conserved”.

5. Line 347. Discussion

Responses: Corrected.

6. Line 376 .The suggestion that "oligomerization uniquely evolved in TaqCad1" is misleading - it is probably conserved in the broader family.

Responses: We revised the statement to “oligomerization uniquely evolved in CARF family of enzymes: to reflect the broad applicability.

7. Line 376. The term "protein network" is imprecise, suggesting multiple proteins.

Responses: We revised the phrase to “a network of interactions with TaqCad1”.

8. Line 382. Interestingly

Responses: Corrected

9. Line 383. "resemble was observed" is not grammatically correct.

Responses: We thank the reviewer for pointing this out. We deleted “resemble”.

10. Line 389. What is meant by "unique physiological context" here?

Responses: We thank the reviewer for pointing this out. We rephrased it to “unique cellular context”.

11. Figure 8. What is "standard deviation of the mean"? Is this SD or SEM? For triplicate data points, SD should be used.

Responses: We thank the reviewer for pointing this out. We revised to SD from SDM.

12. Figure 8C is not really strengthening the paper, since it seems the empty vector control has a clearer phenotype than TaqCad1. Probably worth removing this.

Responses: We agree with the reviewer on this point. The empty vector control possibly reflects the experimental variation while the TaqCad1 response is moderate. We removed this panel and kept the rest of the result in Supplementary Figure 8.

Dear Prof. Li,

Thank you for submitting a revised version of your manuscript. Your study has now been seen by both original referees, who find that their previous concerns have been addressed and now recommend publication of the manuscript. There remain only a few mainly editorial points that have to be addressed before I can extend formal acceptance of the manuscript:

- On the abstract page of the manuscript, please include 4-5 general keyword terms to enhance searchability.
- Please place the "data availability section before the Acknowledgments section
- Please include a "Disclosure and Competing Interests Statement", in accordance with our updated Guide to Authors (<https://www.embopress.org/competing-interests>)
- Please resolve the AUTHORS: name discrepancies - Arthur Carl Whittington in the ms vs. Arthur Whittington in the system; Despoina Trasanidou in the ms vs. Depoina Trasanidou in the system; Carl Raymund P. Salazar in the ms vs. Carl Salazar
- As we are switching from a free-text author contribution statement towards a more formal statement based on Contributor Role Taxonomy (CRediT) terms, please remove the present Author Contribution section and instead specify each author's contribution(s) directly in the Author Information page of our submission system during upload of the final manuscript. See <https://casrai.org/credit/> for more information.
- Please place the REFERENCES section before figure legends
- The FUNDING INFO should be part of the Acknowledgments so separate 'Funding' section heading is not needed
- Please adjust the in-text callouts for individual figures and figure panels: e.g. missing a callout for Fig. 5C
- Please provide suggestions for a short 'blurb' text prefacing and summing up the conceptual aspect of the study in two sentences (max. 250 characters), followed by 3-5 one-sentence 'bullet points' with brief factual statements of key results of the paper; they will form the basis of an editor-written 'Synopsis' accompanying the online version of the article. Please also provide an altered synopsis image, making sure that the aspect ratio conforms to our website's format - it should be exactly 550 pixels wide and between 300-600 pixels high. Please remove the highlights from the manuscript file to e.g. be uploaded separately as bullet points for the synopsis text.
- Please provide the Reagent and Tools Table. For more information, please check <https://www.embopress.org/page/journal/14602075/authorguide#structuredmethods> and download the template for Reagent Table
- Please make sure that you have uploaded all source data: e.g. 1B appears to be missing: Please upload SD as 1 folder per figure
- Movies: 3 movies upladd, they all play; Please remove the legends from the ms and each should be zipped up with its corresponding movie so that we have 3 zip folders upladd Movie EV1-EV3
- Please provide the specific URLs for 9OF1, 9EBT, 9OFB, EMD-619, EMD-47886, EMD-47887, EMD-47888, EMD-47890, EMD-70417, EMD-70419, EMD-70422, EMD-70423, EMD-70424, EMD-70425 datasets in the data availability statement. Please provide also an URL for the PDB validation report.
- Please note that the exact p values are not provided in the legend of figure S3B
- Please note that information related to n is missing in the legends of figures 2A-C; EV1 A-C; S1 D,
- Please note that the error bars are not defined in the legends of figures 2A-C; EV1 A-C; S1 D
- Please note that the measure of center for the error bars needs to be defined in the legend of figure 1B

With best regards,

Cornelius Schneider

Cornelius Schneider, PhD
Editor | The EMBO Journal
c.schneider@embojournal.org

Please refer to our figure preparation guideline in order to ensure proper formatting and readability in print as well as on screen:

See also figure legend guidelines:

<https://www.embopress.org/page/journal/14602075/authorguide#figureformat>

Use the link below to submit your revision:

All editorial and formatting issues were resolved by the authors.

Dear Prof. Li,

I am pleased to inform you that your manuscript has been accepted for publication in the EMBO Journal.

Yours sincerely,

Cornelius Schneider, PhD
Editor
The EMBO Journal
c.schneider@embojournal.org
